# Continual Unlearning for Text-to-Image Diffusion Models: A Regularization Perspective

**Justin Lee**[1*]    **Zheda Mai**[1*]    **Jinsu Yoo**[1]    **Chongyu Fan**[2]
**Cheng Zhang**[3]    **Wei-Lun Chao**[1,4]

[1]The Ohio State University    [2]Michigan State University
[3]Texas A&M University    [4]Boston University

## Abstract

Machine unlearning—the ability to remove designated concepts from a pre-trained model—has advanced rapidly, particularly for text-to-image diffusion models. However, existing methods typically assume that unlearning requests arrive all at once, whereas in practice they often arrive sequentially. We present the first systematic study of **continual unlearning** in text-to-image diffusion models and show that popular unlearning methods suffer from **rapid utility collapse**: after only a few requests, models forget retained knowledge and generate degraded images. We trace this failure to cumulative parameter drift from the pre-training weights and argue that **regularization is crucial** to addressing it. To this end, we study a suite of add-on regularizers that (1) mitigate drift and (2) remain compatible with existing unlearning methods. Beyond generic regularizers, we show that **semantic awareness** is essential for preserving concepts close to the unlearning target, and propose a **gradient-projection method** that constrains parameter drift orthogonal to their subspace. This substantially improves continual unlearning performance and is **complementary** to other regularizers for further gains. Taken together, our study establishes continual unlearning as a fundamental challenge in text-to-image generation and provides insights, baselines, and open directions for advancing safe and accountable generative AI.

**Continual Unlearning of "*Styles*"**    **Continual Unlearning of "*Objects*"**

Figure 1: **Continual unlearning leads to catastrophic degradation.** The pre-trained model (first column) continually unlearns 12 concepts $\mathcal{C}^{\star} = \{c_1^{\star}, c_2^{\star}, \ldots, c_{12}^{\star}\}$ (*e.g.*, Abstractionism or Bears). Different rows display various prompts used for image generation. In each row, red boxes highlight images where some specific concepts have been unlearned. Ideally, images without red boxes should remain conceptually intact. However, as illustrated in the bottom row, continual unlearning significantly impairs the model's ability to retain concepts. Notably, after unlearning 12 concepts (last column), the model fails to generate meaningful content.

---

* Equal Contribution. Project Page: `https://justinhylee135.github.io/CUIG_Project_Page/`

# 1 INTRODUCTION

Recent advances in text-to-image generation, driven primarily by diffusion models (DMs), have achieved unprecedented success in producing high-quality images across diverse concepts (Rombach et al., 2022; Kawar et al., 2023; Zhang et al., 2024a; Nichol et al., 2021). This versatility stems from training on massive, internet-scale datasets, but such broad data collection introduces serious ethical and legal risks: models may reproduce copyrighted material, generate harmful or biased content, and perpetuate stereotypes (Schramowski et al., 2023; Vinker et al., 2023). In response, regulations such as CCPA (California Attorney General) now grant individuals the right to request removal of their personal or copyrighted content. However, retraining large DMs from scratch for every request is computationally infeasible—for example, retraining Stable Diffusion v2 on LAION-5B (Schuhmann et al., 2022) requires roughly 150,000 GPU-hours (Gandikota et al., 2023a). As a result, *machine unlearning* has emerged as a practical alternative, aiming to selectively erase undesired generative capabilities (*e.g.*, a person's likeness or an artistic style) from pre-trained models without full retraining (Hong et al., 2024; Gandikota et al., 2023a; Kumari et al., 2023).

Despite notable progress in unlearning for DMs, most methods assume that unlearning requests arrive simultaneously (Wu et al., 2025b; Gandikota et al., 2023a; Kumari et al., 2023; Wu et al., 2025a). In reality, such requests are typically sequential—for example, a parent may request the removal of violent concepts one day, followed later by an artist seeking the exclusion of copyrighted artworks.

To reflect this real-world setting, we introduce ***Continual Unlearning (CU)*** for text-to-image generation, defined as the sequential removal of targeted generative capabilities subject to three requirements: (i) effective erasure of newly targeted concepts, (ii) preservation of prior unlearning, and (iii) retention of all unrelated generative abilities (Figure 2). While CU has recently been studied in large language models (LLMs) (Chen & Yang, 2023; Jang et al., 2022), it remains largely unexplored in image generation. We fill this gap with the **first comprehensive empirical study of CU for text-to-image diffusion models**, and introduce a benchmark that extends UNLEARNCANVAS (Zhang et al., 2024b) with style- and object-level unlearning sequences (Figure 1). We outline major insights as follows.

- **Continual unlearning suffers rapid utility collapse.** Popular unlearning methods (Kumari et al., 2023), while effective for removing one or a few concepts simultaneously, break down in the continual setting. After only a handful of requests, models forget retained knowledge and produce degraded images even for unrelated concepts. Our analysis attributes this failure to *cumulative parameter drift*, as successive unlearning steps push the model farther from its pre-training manifold. Consistently, we observe much larger parameter shifts in continually unlearned models than in those where all target concepts are unlearned simultaneously or independently.
- **Generic add-on regularizers partially alleviate collapse.** Motivated by the above, we explore regularizers that can be seamlessly integrated into existing unlearning methods to mitigate drift. These include (i) constraining the update norm relative to previously unlearned models, (ii) selectively updating parameters most critical for the target concepts, and (iii) merging independently unlearned models. These approaches reduce drift and improve preservation of concepts across domains (*e.g.*, generating objects after unlearning styles).
- **Semantic awareness is crucial for in-domain retention.** Retaining *in-domain* capabilities (*e.g.*, unlearning one style while preserving others) remains highly challenging, often leading to sharp utility drops even with regularizers. Empirically, we find a strong negative correlation between retention performance and the text-embedding similarity of the retention concept to the unlearning concept (see Figure 7a), underscoring the need for semantic awareness.
- **Gradient projection provides a principled solution.** We propose a **gradient-projection method** that imposes a hard constraint on parameter updates, forcing them to be orthogonal to the subspace spanned by semantically close concepts. This minimizes unintended interference, substantially improves in-domain retention, and remains *complementary* to other regularizers for further gains.

**Remark.** Rather than proposing a new continual unlearning algorithm, we focus on developing compatible solutions that enhance existing methods—an approach we believe will have a broader impact. Interestingly, the regularizers we study are also effective for unlearning single concepts, the standard setting in the unlearning literature. Their benefits, however, are most pronounced in the continual scenario, particularly as the sequence length grows. Overall, our study provides robust reference points for advancing continual unlearning, underscoring its challenges, opportunities, and promising directions for future work.

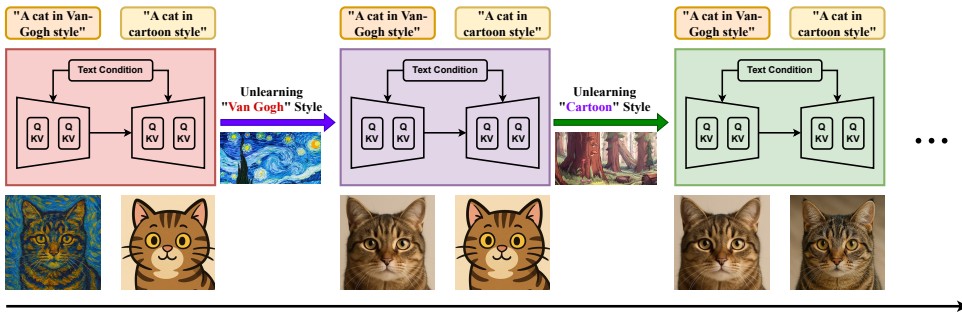

**Figure 2: The *ideal outcomes* in continual unlearning.** The pre-trained model continually unlearns two styles. Given the prompts to generate a cat in "Van Gogh" and "Cartoon" styles, the generated images should accurately reflect the styles. After the first unlearning step, the image for "Cartoon" should remain conceptually unchanged, while the image for "Van Gogh" should no longer exhibit the "Van Gogh" style. Following the second unlearning step, both the "Van Gogh" and "Cartoon" should be removed, while the concept "cat" should be retained.

## 2 RELATED WORK

**Machine Unlearning in Diffusion Models.** Diffusion models revolutionized image generation by training on internet-scale data (Schuhmann et al., 2022; Wang et al., 2025). However, reliance on such data introduces risks of harmful outputs, copyright violations, and biases (Schramowski et al., 2023; Zhang et al., 2026). Unlearning aims to remove undesirable generative capabilities without retraining from scratch (Hong et al., 2024; Gandikota et al., 2023a; Wu et al., 2025a). Widely adopted methods like ConAbl (Kumari et al., 2023) map unlearning concepts to benign anchors, while recently proposed SculpMem (Li et al., 2025) improves ConAbl with a dynamic mask. Nevertheless, most methods still assume unlearning requests arrive simultaneously, overlooking realistic scenarios where requests arrive sequentially. Our study addresses the unexplored question: *Are unlearning methods still effective in continual settings, and how can they be adapted to unlearn continually?*

**Continual Unlearning.** Continual Unlearning (CU) is an emerging direction where removal requests arrive sequentially rather than all at once. CU was first studied in LLMs, aiming to unlearn user-sensitive knowledge or undesired capabilities while preserving general language ability (Chen & Yang, 2023; Gao et al., 2024; Jang et al., 2022). In contrast, CU for image generation remains largely unexplored. We address this gap with a systematic study of CU for text-to-image diffusion models, diagnosing utility collapse and proposing mitigation strategies.

**Continual Learning.** Continual unlearning and continual learning Mai et al. (2022; 2026) are closely related: both update existing models while striving to preserve acquired capabilities. Unlike continual learning, where the model aims to learn new concepts, both the concepts to be removed and retained are *already known* by the model in continual unlearning, amplifying interference risks. Despite this fundamental difference, principles from continual learning remain highly relevant (Heng & Soh, 2023). Motivated by weight and gradient-based regularization and selective fine-tuning (Zenke et al., 2017; Mazumder et al., 2021; Lopez-Paz & Ranzato, 2017), we investigate whether these mechanisms can be repurposed to enable effective unlearning without utility collapse. By bridging insights from continual learning to continual unlearning, we set the stage for future investigations.

**Detailed Related Work.** Due to the page limit, we include detailed related work in Appendix F.

## 3 PRELIMINARY

### 3.1 MACHINE UNLEARNING FOR TEXT-TO-IMAGE DIFFUSION MODELS

Diffusion models (DMs) generate images by progressively denoising an initial Gaussian sample. At each step $t$, a neural network $\epsilon_{\boldsymbol{\theta}^\dagger}$ estimates the noise component in the current state $\boldsymbol{x}_t$, producing a cleaner state $\boldsymbol{x}_{t-1}$. Iterating this process yields $I = \boldsymbol{x}_0$, the final image. For text-to-image generation, a text prompt $q$ is additionally input to $\epsilon_{\boldsymbol{\theta}^\dagger}$ to guide the denoising trajectory, *i.e.*, $\epsilon_{\boldsymbol{\theta}^\dagger}(\boldsymbol{x}_t, q, t)$. We denote the full generation process by $G_{\boldsymbol{\theta}^\dagger}$, with output image $I = G_{\boldsymbol{\theta}^\dagger}(q)$.

Ideally, if a prompt $q$ contains a concept $c$ (*e.g.*, an art style or object), the generated image $I = G_{\theta^\dagger}(q)$ should accurately reflect it. This can be evaluated with a recognition model $F$, such as CLIP (Radford et al., 2021), by checking whether the predicted label $\hat{c} = F(I)$ satisfies $\hat{c} = c$.

Unlearning aims to update the pre-trained model weights $\theta^\dagger$ so as to remove the generative ability for designated target concepts. Let $c^\star$ denote a target concept, and let $\theta^\star$ denote the model parameters after unlearning $c^\star$. For any prompt $q$ containing $c^\star$, the generated image $I = G_{\theta^\star}(q)$ should satisfy $F(I) \neq c^\star$. For all other concepts $c \neq c^\star$, the model should retain them; that is, if $c$ appears in the prompt $q$, then we should have $F(G_{\theta^\star}(q)) = c$.

## 3.2 PAPER STRUCTURE

The goal of this paper is to introduce, analyze, and improve continual unlearning (CU). We structure the remainder as follows: section 4 defines the CU setting and presents our benchmark; section 5 evaluates baseline CU approaches, identifies their limitations, and investigates the root cause of failure; section 6 studies generic regularizers as a remedy, while section 7 demonstrates the importance of semantic-aware regularizers for preserving in-domain generative capabilities. Finally, section 8 provides further analysis of the unlearning dynamics, offering insights for future CU methods.

## 4 CONTINUAL UNLEARNING: SETUP AND BENCHMARK

### 4.1 SETUP

**Motivation.** In practice, a model may be asked to erase multiple concepts $\mathcal{C}^\star = \{c_1^\star, c_2^\star, \ldots, c_N^\star\}$. If all requests arrive at once, one can update $\theta^\dagger$ to jointly unlearn all $c^\star \in \mathcal{C}^\star$. In reality, however, requests typically arrive sequentially, calling for *continual unlearning* (CU) methods that remove each concept as it is received.

**Definition.** Without loss of generality, assume requests arrive in order $c_1^\star, \ldots, c_N^\star$. Let $\theta_n^\star$ denote the model obtained after unlearning the first $n$ concepts. For any concept $c$ appearing in a prompt $q$, the model should satisfy:

$$F(G_{\theta_n^\star}(q)) = \begin{cases} \neq c, & \text{if } c \in \{c_1^\star, \ldots, c_n^\star\}; \quad \text{(unlearned)} \\ = c, & \text{otherwise.} \qquad\qquad \text{(retained)} \end{cases}$$

**Metrics.** Following UNLEARNCANVAS we evaluate CU after the $n$-th request with two metrics:

- **Unlearning Accuracy (UA).** For each unlearned concept $c \in \{c_1^\star, \ldots, c_n^\star\}$, we count success when $F$ does *not* return $c$ for an image generated from a prompt containing it, *i.e.*, $F(G_{\theta_n^\star}(q)) \neq c$. UA is the fraction of successes across generated images.
- **Retention Accuracy (RA).** For each retained concept $c \notin \{c_1^\star, \ldots, c_n^\star\}$, we count success when $F$ correctly returns $c$ for an image generated from a prompt containing it, *i.e.*, $F(G_{\theta_n^\star}(q)) = c$.

To better analyze retention, we partition concepts into two subsets: an *in-domain* set, containing those semantically or structurally related to the unlearned concepts, and a *cross-domain* set for the rest. For example, if the unlearning targets image styles (*e.g.*, "Cartoon"), then other styles (*e.g.*, "Van Gogh") are in-domain, while objects (*e.g.*, "Cat") are cross-domain. Accordingly, we report **In-Domain Retention Accuracy (RA-I)** and **Cross-Domain Retention Accuracy (RA-C)**.

### 4.2 BENCHMARK

**Data and Model Source.** Prior works on concept unlearning have lacked standardized evaluation protocols, relying on heterogeneous metrics such as CLIP Score similarity (Wu et al., 2025b; Kumari et al., 2023; Gandikota et al., 2023a) or subjective human evaluation (Gandikota et al., 2023a), thereby hindering fair comparison.

To address this, we adopt UNLEARNCANVAS (Zhang et al., 2024b) as our evaluation backbone. It provides a fine-tuned Stable Diffusion (Rombach et al., 2022) checkpoint $G_{\theta^\dagger}$ and specialized classifiers $F$ trained to recognize 60 artistic styles and 20 object categories. The DM checkpoint ensures that all 80 concepts can be generated with high accuracy (>98% top-1), while the classifiers offer a standardized and objective means of reporting **UA**, **RA-I**, and **RA-C**.

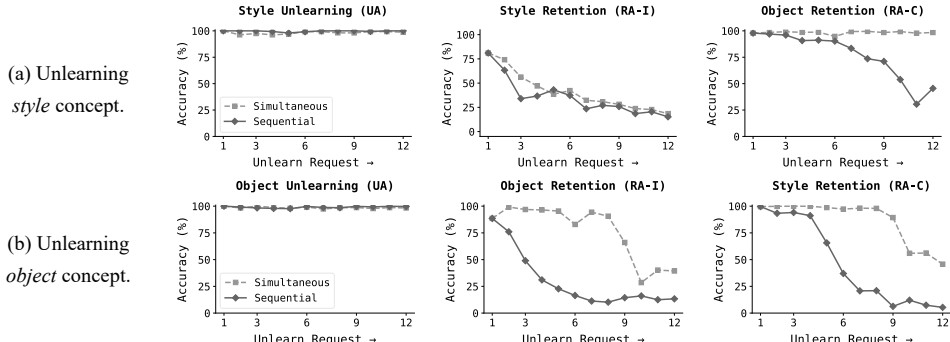

Figure 3: ConAbl (Kumari et al., 2023) fails when unlearn requests arrive continually. Although it performs well at the initial request, unlearning sequentially leads to poor retention. Simultaneously unlearning all requests better preserves retention, but comes with a much higher cost. Plots for SculpMem (Li et al., 2025) in Appendix A.

**Evaluation Protocol.** To systematically evaluate CU performance, we consider two settings for constructing the unlearning targets $\mathcal{C}^\star = \{c_1^\star, c_2^\star, \ldots, c_N^\star\}$:

- **Continual Style Unlearning.** We sample a random sequence of 12 unique artistic styles to be unlearned. To evaluate retention, we hold out 12 additional styles and 8 objects that are never targeted during unlearning. This allows us to measure both in-domain retention (other styles) and cross-domain retention (objects).
- **Continual Object Unlearning.** Symmetrically, we sample a random sequence of 12 unique objects to be unlearned. The same held-out evaluation set from the style setting is used, ensuring fair comparison across settings without biases from different evaluation sets.

After each unlearning request, we follow UNLEARNCANVAS (Zhang et al., 2024b) to generate diverse images for both unlearned and retained concepts, using the template "A {object} image in {style} style." For example, after erasing the "Van Gogh" style, we generate 5 images (different random seeds) for each of the 8 held-out objects conditioned on this style to compute UA. In total, this yields 40 images per style concept ($5 \times 8$). Analogously, when unlearning an object concept, we generate 5 images for each of the 12 held-out styles, yielding 60 images per object concept ($5 \times 12$).

## 5 CONTINUAL UNLEARNING SUFFERS RAPID UTILITY COLLAPSE

### 5.1 EXISTING METHODS FAIL TO UNLEARN CONTINUALLY

**Unlearning Methods.** We first examine how existing methods behave in a continual setting, focusing on two representative ones: the widely adopted Concept Ablation (ConAbl) (Kumari et al., 2023) and the recently proposed SculpMem (Li et al., 2025). Like many unlearning methods, both define an unlearning loss $\mathcal{L}_{\text{unlearn}}(\boldsymbol{\theta}, \mathcal{C})$ that depends on the model parameters $\boldsymbol{\theta}$ and the target concept set $\mathcal{C}$. Minimizing this loss with initialization $\boldsymbol{\theta}^\dagger$ (the pre-trained weights) yields an unlearned model $\boldsymbol{\theta}^\star$.

**Extension to CU.** We adapt these methods to continual unlearning using two strategies:

- **Sequential:** At the $n$-th request, the model is incrementally updated by minimizing $\mathcal{L}_{\text{unlearn}}(\boldsymbol{\theta}, \{c_n^\star\})$, starting from the previously unlearned model $\boldsymbol{\theta}_{n-1}^\star$.
- **Simultaneous:** At the $n$-th request, the model is retrained from the pre-trained weights $\boldsymbol{\theta}^\dagger$ to jointly unlearn all target concepts so far, *i.e.*, minimizing $\mathcal{L}_{\text{unlearn}}(\boldsymbol{\theta}, \{c_1^\star, \ldots, c_n^\star\})$.

**Results.** Both ConAbl and SculpMem perform well for single-concept unlearning, achieving high UA, RA-I, and RA-C on the first request (Figure 3). However, as additional concepts are unlearned sequentially, their utility collapses: while UA remains high, the models rapidly lose the ability to generate unrelated concepts, leading to drastic drops in RA-I and RA-C (Figure 3; Figure 1).

By contrast, the simultaneous strategy preserves utility more effectively, but at a prohibitive cost: each new request requires *re-unlearning* all prior concepts from scratch, making training time grow with the total number of requests (Appendix E). This efficiency-utility trade-off underscores the need for continual unlearning methods that can handle sequential requests without collapsing retention.

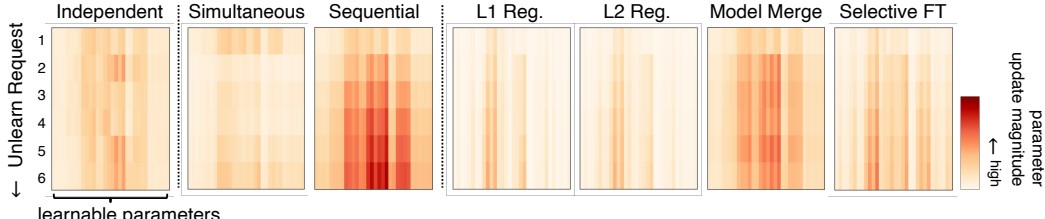

Figure 4: ConAbl's (Kumari et al., 2023) cumulative $\ell_2$ parameter drift w.r.t the pre-trained model. Sequential unlearning exhibits severe cumulative drift with more unlearned concepts compared to simultaneous unlearning. Our add-on regularizers effectively mitigate this drift and demonstrate better retention (Figure 6).

## 5.2 WHY DOES SEQUENTIAL UNLEARNING FAIL?

**Empirical Observations.** The above results raise an important question. Existing methods can unlearn multiple concepts with high retention when applied simultaneously, yet their effectiveness collapses when applied sequentially. To understand this discrepancy, we analyze the unlearned models $\boldsymbol{\theta}_n^\star$ after the $n$-th request under both strategies, focusing on their deviation from the pre-trained weights $\boldsymbol{\theta}^\dagger$. As shown in Figure 4, after the first request, both strategies exhibit a similar degree of parameter drift, measured by $\|\boldsymbol{\theta}_n^\star - \boldsymbol{\theta}^\dagger\|_2$. With more requests, however, drift grows dramatically under sequential unlearning while remaining nearly constant under the simultaneous strategy. For comparison, we also unlearn each concept *independently* from $\boldsymbol{\theta}^\dagger$. The norms of these parameter shifts remain similar to those from simultaneous unlearning, despite the latter involving progressively more concepts.

These findings suggest the following hypothesis: *High retention in sequential continual unlearning requires regularizing parameter drift.*

**Theoretical Analysis.** We seek to provide a theoretical perspective on the empirical findings. Intuitively, the pre-trained weights $\boldsymbol{\theta}^\dagger$ encode the model's original generative capabilities. Therefore, keeping the unlearned model $\boldsymbol{\theta}^\star$ close to $\boldsymbol{\theta}^\dagger$ should help preserve these capabilities.

Building on the loss approximation framework from continual learning (Yin et al., 2020; Zenke et al., 2017; Aljundi et al., 2018), we formalize this intuition using a Taylor expansion of the retention loss $L$ around $\boldsymbol{\theta}^\dagger$ (full derivation in Appendix B). This yields the following bound on the change in $L$:

$$|L(\boldsymbol{\theta}^\star, \mathcal{C}^r) - L(\boldsymbol{\theta}^\dagger, \mathcal{C}^r)| \leq \|\nabla L(\boldsymbol{\theta}^\dagger, \mathcal{C}^r)\| \cdot \|\boldsymbol{\theta}^\star - \boldsymbol{\theta}^\dagger\| + \tfrac{1}{2}\|H(\boldsymbol{\theta}^\dagger, \mathcal{C}^r)\| \cdot \|\boldsymbol{\theta}^\star - \boldsymbol{\theta}^\dagger\|^2,$$

where $\boldsymbol{\theta}^\dagger$ is the pre-trained model, $\boldsymbol{\theta}^\star$ the unlearned model, $\mathcal{C}^r$ the retention set, and $H$ the Hessian.

This inequality shows that the change in retention loss is Lipschitz continuous w.r.t the parameter update, meaning the loss grows proportionally (up to a constant) to $\|\boldsymbol{\theta}^\star - \boldsymbol{\theta}^\dagger\|$. Hence, preserving utility depends directly on how close the unlearned model remains to the pre-trained parameters.

Moreover, when the gradient and Hessian terms are small—typically the case near the optimum $\boldsymbol{\theta}^\dagger$—the update norm becomes the dominant factor. To validate this, we estimate the curvature of the retention loss by perturbing the pre-trained weights and measuring the ratio of gradient change (evaluated on the UNLEARNCANVAS training set) to weight perturbation. The estimated Hessian coefficients are minuscule, confirming that the retention loss lies in a smooth basin (see Appendix B).

## 6 ADD-ON REGULARIZATION FOR SEQUENTIAL CONTINUAL UNLEARNING

Motivated by our empirical and theoretical analyses, we explore add-on regularization strategies that constrain parameter drift to improve retention. These approaches differ in how they measure drift (*e.g.*, using different norms) and how they impose the constraint (overview in Figure 5).

### 6.1 UPDATE NORM REGULARIZATION

We begin with the most common approach: directly penalizing the norm of the parameter update. At the $n$-th request, we augment the unlearning loss with a regularization term:

$$\mathcal{L}_{\text{unlearn}}(\boldsymbol{\theta}, \{c_n^\star\}) + \lambda \|\boldsymbol{\theta} - \boldsymbol{\theta}_{n-1}^\star\|_p^p,$$

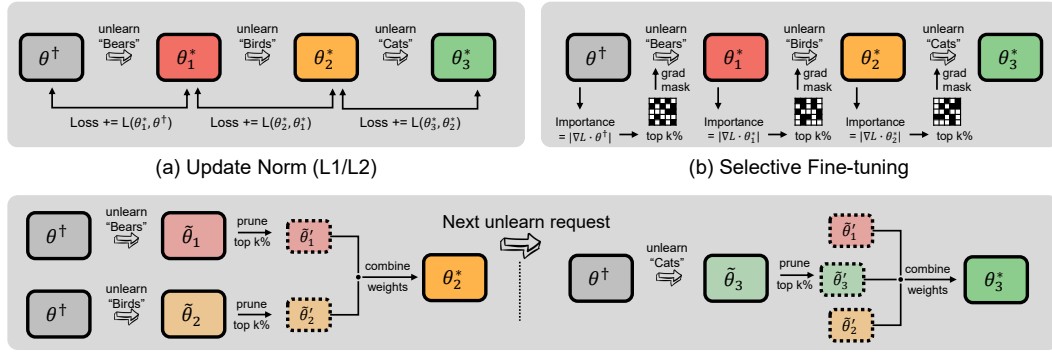

Figure 5: Overview of our add-on regularizers. (a) L1/L2 penalizes the norm of the parameter update relative to the previous checkpoint. (b) Selective Fine-tuning restricts updates to the top-$k$% most important parameters. (c) Model merging unlearns each concept independently and combines the resulting models.

where $\boldsymbol{\theta}^\star_{n-1}$ is the model obtained after the $(n-1)$-th request and serves as the initialization for $\boldsymbol{\theta}$. We consider two choices of $p$: the $L_1$ norm, which encourages sparse updates, and the $L_2$ norm, which distributes the update across parameters, preventing any single weight from drifting excessively.

## 6.2 SELECTIVE FINE-TUNING (SELFT)

Inspired by work in continual learning (Wang et al., 2024), unlearning (Fan et al., 2023), and model pruning (Wang et al., 2020; Cheng et al., 2024), we investigate Selective Fine-tuning (SelFT) as an alternative to norm-based regularization. Unlike the $L_1$ penalty, which encourages isotropic sparsity, SelFT explicitly restricts updates to parameters deemed critical for the unlearning loss.

While SelFT is often used as an umbrella term for gradient-masking or saliency-based approaches, in this paper, we adopt the method by Nguyen et al. (2024). At the $n$-th request, given $\boldsymbol{\theta}^\star_{n-1}$, we compute parameter importance in a single forward pass using a first-order Taylor approximation:

$$\text{Importance}(d) = \left| \nabla_{\theta[d]} \mathcal{L}_{\text{unlearn}}(\boldsymbol{\theta}^\star_{n-1}, \{c^\star_n\}) \cdot \theta^\star_{n-1}[d] \right|.$$

We then select the top $k$% most important parameters and update only those during unlearning. By explicitly limiting the number of tunable parameters, SelFT constrains drift while still allowing effective concept removal.

## 6.3 MODEL MERGE

Since independently unlearned models for each concept remain close to the pre-trained weights $\boldsymbol{\theta}^\dagger$ (Figure 4), we investigate model merging (Yang et al., 2024) to integrate their effects while staying near the original model. As all such models originate from the same checkpoint, they are likely to lie in the same loss basin (Frankle et al., 2020). Interpolating within this basin keeps retention loss low, allowing merged models to preserve utility while handling multiple unlearning requests.

Concretely, let $\tilde{\boldsymbol{\theta}}_n$ denote the $n$-th independently unlearned model. We adopt TIES-Merging (Yadav et al., 2023) to construct $\boldsymbol{\theta}^\star_n$ by merging $\tilde{\boldsymbol{\theta}}_1, \ldots, \tilde{\boldsymbol{\theta}}_n$. TIES first *prunes* each $\tilde{\boldsymbol{\theta}}_n$ by retaining the top-$k$% parameter updates (ranked by absolute deviation from $\boldsymbol{\theta}^\dagger$), yielding a pruned model $\tilde{\boldsymbol{\theta}}'_n$, and then *merges* them by averaging. Importantly, merging imposes a form of regularization: the merged model $\boldsymbol{\theta}^\star_n$ lies in the affine hull of the pruned models, which restricts updates to the subspace spanned by $\{(\tilde{\boldsymbol{\theta}}'_1 - \boldsymbol{\theta}^\dagger), \ldots, (\tilde{\boldsymbol{\theta}}'_n - \boldsymbol{\theta}^\dagger)\}$.

## 6.4 RESULTS AND INSIGHTS

As shown in Figure 4, add-on regularizers substantially reduce parameter drift during sequential unlearning, yielding clear improvements in both RA-I and RA-C (Figure 6). Among them, model merging delivers the strongest overall retention.

However, in-domain retention (RA-I) remains particularly challenging: across all regularizers, RA-C consistently surpasses RA-I. This gap is expected, as concepts within the same domain are

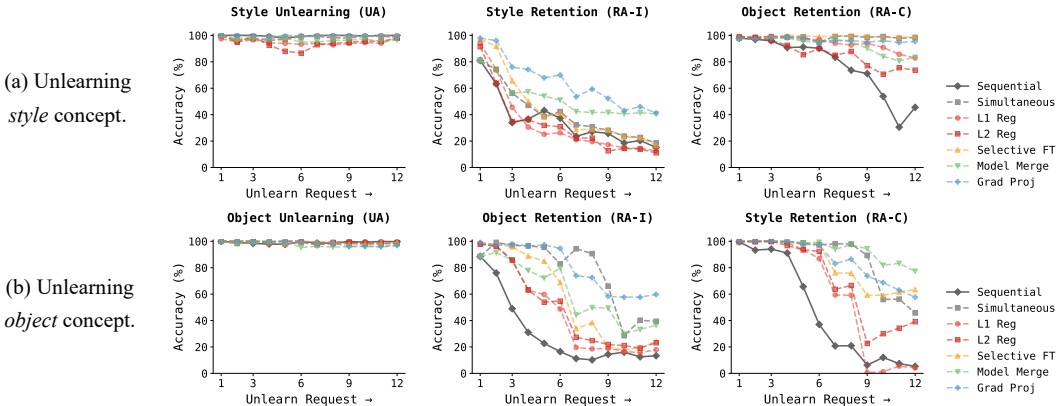

Figure 6: Add-on regularizers substantially improve retention under sequential unlearning with ConAbl, yielding particularly strong gains in cross-domain retention (RA-C). By removing gradient components that interfere with semantically related concepts, our gradient projection method further improves in-domain retention (RA-I). Results for SculpMem are provided in Appendix A.

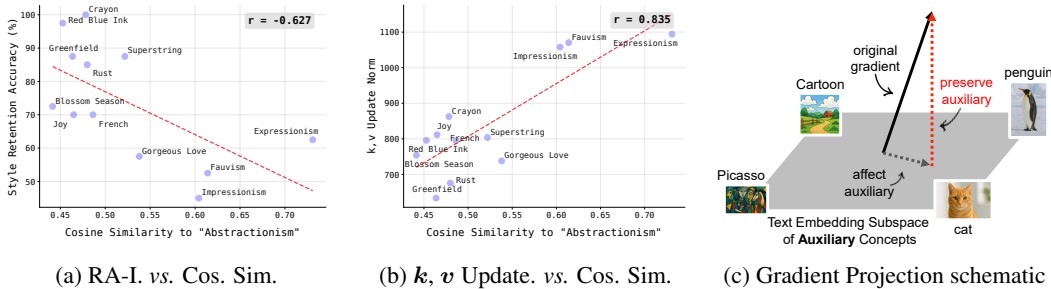

(a) RA-I. *vs.* Cos. Sim.   (b) $k$, $v$ Update. *vs.* Cos. Sim.   (c) Gradient Projection schematic

Figure 7: (a) RA-I decreases as text-embedding cosine similarity to the unlearned concept (*e.g.*, "Abstractionism" style) increases (strong negative correlation). (b) Change in $k,v$ grows with text-embedding cosine similarity (strong positive correlation). (c) These trends motivate our gradient-projection method, which removes gradient components that alter $k$, $v$ for semantically similar (auxiliary) concepts.

semantically closer (*e.g.*, "Bear" vs. "Cat") than cross-domain pairs (*e.g.*, "Bear" vs. "Van Gogh"), making them more prone to interference during unlearning.

## 7 GRADIENT PROJECTION FOR SEMANTIC-AWARE CONTINUAL UNLEARNING

### 7.1 SEMANTIC AWARENESS IS CRUCIAL

To systematically quantify the link between semantic similarity and retention difficulty, we unlearn the style concept "Abstractionism" and measure the retention accuracy of other style concepts, alongside their cosine similarity to the text embedding of "Abstractionism." As shown in Figure 7a, retention accuracy exhibits a strong negative correlation with embedding similarity to the unlearned concept, underscoring the need for semantic awareness in continual unlearning. In particular, special care is required to preserve generative capabilities that are semantically close to the unlearning target.

### 7.2 PROJECTION MATRICES IN CROSS-ATTENTION AS THE KEY SOURCE OF INTERFERENCE

To design a semantic-aware strategy for unlearning in diffusion models, we first need to understand how semantically close concepts interfere with each other in the model. The standard way to inject a prompt $q$ into the denoising network $\epsilon_{\boldsymbol{\theta}^\dagger}(\boldsymbol{x}_t, q, t)$ (cf. subsection 3.1) is through cross-attention, where the current latent state $\boldsymbol{x}_t$ serves as queries to attend to the token embeddings of the text prompt $q$. In this mechanism, the prompt tokens are projected into keys and values that $\boldsymbol{x}_t$ retrieves from.

For simplicity, assume each concept $c$ corresponds to a single token, with embedding $E(c)$. Its key and value vectors are then given by

$$\boldsymbol{k} = \boldsymbol{W}_K E(c), \quad \boldsymbol{v} = \boldsymbol{W}_V E(c),$$

where $\boldsymbol{W}_K$ and $\boldsymbol{W}_V$ are learnable projection matrices. Unlearning a target concept $c^\star$ amounts to updating $\boldsymbol{W}_K$ and $\boldsymbol{W}_V$ so that $E(c^\star)$ is mapped far away from its original $(\boldsymbol{k}, \boldsymbol{v})$, making it inaccessible for $\boldsymbol{x}_t$ to retrieve during generation.

However, because linear projections approximately preserve neighborhood structure, semantically similar concepts $c$ and $c^\star$ remain close after projection:

$$\|\boldsymbol{A}E(c) - \boldsymbol{A}E(c^\star)\| \ \leq \ \|\boldsymbol{A}\| \cdot \|E(c) - E(c^\star)\|,$$

for any linear operator $\boldsymbol{A}$. Consequently, updating $\boldsymbol{W}_K$ and $\boldsymbol{W}_V$ to suppress $c^\star$ inevitably distorts the embeddings of nearby concepts $c$ as well. Empirically, we observe that higher text-embedding similarity indeed correlates with greater distortion in $(\boldsymbol{k}, \boldsymbol{v})$ (Figure 7b).

## 7.3 GRADIENT PROJECTION TO SUPPRESS INTERFERENCE

We propose a method to suppress the undesired influence of updating $\boldsymbol{W}_K$ and $\boldsymbol{W}_V$ on semantically similar concepts. After obtaining the *unlearning gradients* with respect to the projection matrices,

$$\nabla_{\boldsymbol{W}_K} \mathcal{L}_{\text{unlearn}}(\boldsymbol{\theta}, \{c^\star\}), \quad \nabla_{\boldsymbol{W}_V} \mathcal{L}_{\text{unlearn}}(\boldsymbol{\theta}, \{c^\star\}),$$

we project out the components that perturb nearby concepts to first order (Figure 7c).

Concretely, let $\{c_i\}_{i=1}^M$ denote $M$ auxiliary[1] concepts generated by an LLM and filtered by text-embedding similarity to the target $c^\star$, and let $C = [E(c_1), E(c_2), \ldots, E(c_M)]$ be their embeddings. The span $\mathcal{S} := \text{span}(C)$ approximates the subspace of embedding directions corresponding to semantically similar concepts. We remove gradient components lying in $\mathcal{S}$, ensuring that updates suppress $c^\star$ while minimally distorting its neighbors.

**Details.** Given a vector $g$, its Euclidean projection onto $\mathcal{S}$ is $\text{proj}_{\mathcal{S}}(g) := \arg\min_{u \in \mathcal{S}} \|g - u\|_2^2$. Since any $u \in \mathcal{S}$ can be expressed as $u = C\alpha$ for some $\alpha \in \mathbb{R}^M$, this reduces to a least-squares problem with a closed-form solution $\hat{u} = C\hat{\alpha} = P_{\mathcal{S}} g$, where $P_{\mathcal{S}} := C(C^\top C)^{-1}C^\top$. Here, $P_{\mathcal{S}}$ is the orthogonal projector onto the subspace spanned by $C$, and the complementary projector $P_{\mathcal{S}^\perp} := I - P_{\mathcal{S}}$ removes all components of $g$ that lie in $\mathcal{S}$.

Let $g^\star$ denote the gradient for unlearning the target concept $c^\star$. To suppress interference with semantically similar concepts, we project $g^\star$ onto the orthogonal complement of $\text{span}(C)$ (Figure 7c):

$$g' \ = \ P_{\mathcal{S}^\perp} g^\star \ = \ (I - P_{\mathcal{S}}) g^\star.$$

Here, $g'$ is the projected gradient, which preserves directions useful for unlearning $c^\star$ while eliminating components aligned with nearby concepts. We note that $g' = (I - P_{\mathcal{S}})g'$.

**Lemma 7.1** (First-order invariance). *For any $c \in \text{span}(C)$, the update direction $g'$ produces zero first-order change: $g'^\top c = 0$.*

*Proof.* If $c \in \text{span}(C)$, then $P_{\mathcal{S}} c = c$. Hence $g'^\top c = g'^\top (I - P_{\mathcal{S}})^\top c = g'^\top (c - c) = 0$. □

**Results.** As shown in Figure 6, our gradient-projection method achieves the highest retention accuracy on in-domain concepts (RA-I) across both style and object unlearning settings. However, its retention accuracy on cross-domain concepts (RA-C) is slightly lower than that of SelFT and model merging in the object unlearning case. To address this gap, we examine whether combining our method with existing add-on regularizers can improve RA-C while preserving RA-I. As shown in Figure 8, gradient projection is indeed **compatible** with these add-ons, and their combination yields further improvements.

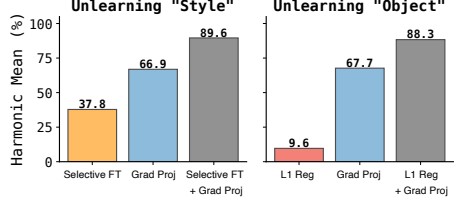

Figure 8: Gradient-projection is compatible with add-on regularizers, yielding additional gains when combining them. The y-axis shows the harmonic mean of UA, RA-I, and RA-C, using ConAbl for unlearning.

---

[1] We define *auxiliary* concepts as those that are semantically related to the target concept but should be retained during unlearning. Our method does not require access to the retain set.

## 8 UNDERSTANDING THE UNLEARNING PROCESS

To complement our previous analysis, we investigate the mechanics of how unlearning modifies the model, revealing insights that may inform the design of future unlearning and regularization methods.

**Unlearning is About Learning.** For anchor-based unlearning methods (Kumari et al., 2023), we find that the parameter updates are driven primarily by the chosen anchor rather than the target being unlearned. Distinct targets mapped to the same anchor induce highly correlated parameter updates (Figure 9). This suggests that anchor-based unlearning functions as representation replacement: the anchor representation is relearned to overwrite the target. This exposes a tradeoff: mapping many targets to a shared anchor may localize parameter modifications, while mapping targets to distinct anchors may disperse updates more broadly across the model.

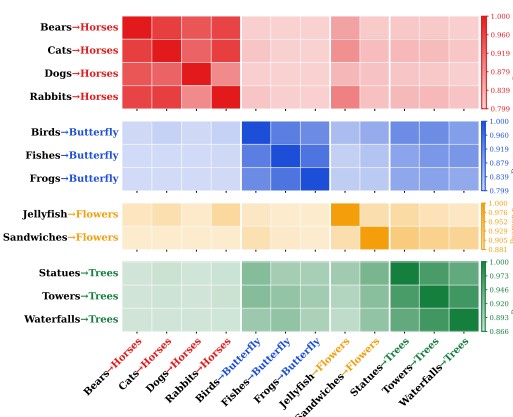

Figure 9: Parameter updates during unlearning are governed by the anchor concept rather than the specific target being removed. Models that share an anchor display highly correlated updates.

**Concept Erasure is All-or-Nothing.** When interpolating between the pre-trained and unlearned model, we find that for many concepts, generated outputs remain visually unchanged across a wide range of interpolation coefficients until a critical threshold is crossed, at which point the target concept is abruptly suppressed (Appendix C). Erasure thus behaves as a sharp transition rather than a gradual process. This complicates model merging, as interpolation coefficients must be carefully chosen to keep every concept above its respective erasure threshold.

**Parameter Drift is Intrinsic to Sequential Unlearning.** One might hypothesize that the greater parameter drift in sequential unlearning (Section 5.2) is simply an artifact of using more optimization steps. To test this, we apply early stopping to both sequential and simultaneous unlearning, terminating each when 99% unlearning accuracy is reached. We then continue unlearning additional concepts until the cumulative optimization steps of simultaneous unlearning match or exceed those of sequential unlearning. Even with comparable totals (*e.g.*, 2,100 each after six concepts), sequential unlearning still accumulates substantially more drift (Appendix D). Understanding what property of simultaneous unlearning keeps drift low despite comparable optimization steps may guide the design of regularizers that bring the same benefit to sequential methods.

## 9 CONCLUSION

We present the first systematic study of continual unlearning for image generation, reflecting real-world scenarios where unlearning requests arrive sequentially. We find that existing methods quickly degrade in utility—forgetting retained concepts and generating low-quality images—and trace this failure to cumulative parameter drift and semantic interference. We show that simple, plug-and-play regularizers based on update norm, selective fine-tuning, model merging, and semantic-aware gradient-projection can substantially restore performance. **For practitioners**, our results suggest that combining selective fine-tuning with gradient-projection provides a strong starting point, effectively constraining both parameter drift and semantic interference to deliver robust retention across in-domain and cross-domain settings.

**Future work.** Understanding how robustness to adversarial recovery attacks evolves across sequential unlearning steps is critical for safe deployment, particularly as it remains unclear whether these challenges compound differently across architectures (*e.g.*, DiT), training objectives (*e.g.*, flow matching), and modalities (*e.g.*, video, speech) beyond diffusion-based image generation. While our plug-and-play regularizers provide a strong foundation, designing natively sequential unlearning methods that anticipate future requests and account for their interactions is a natural next step toward further advancing continual unlearning.

ACKNOWLEDGMENTS

This research is supported by grants from the National Science Foundation (ICICLE: OAC-2112606). We are grateful for the support of the Ohio Supercomputer Center for providing computational resources.

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

APPENDIX

**Disclosure of LLM Usage.** Portions of this manuscript were polished for clarity and readability using an LLM. The LLM was not used to generate research ideas, design experiments, analyze data, or draw conclusions. All scientific content, methods, and results are the authors' original work.

**Appendix Structure.** This appendix is organized as follows. Appendix A provides extended experimental results showing generalized findings across different unlearning methods and erasure settings. Appendix B includes extended theoretical support and additional empirical evidence clarifying the importance of constraining parameter drift. Additional analysis regarding the unlearning process can be found Appendix C, Appendix D, and Appendix E. Finally, detailed related work can be found in Appendix F.

## A   EXTENDED EXPERIMENTAL RESULTS

### A.1   ADDITIONAL RESULTS ON SCULPTING MEMORY

We further validate our findings on Sculpting Memory (SculpMem) (Li et al., 2025), a recent unlearning method designed for multi-concept unlearning. Despite using a dynamic gradient mask and achieving a much stronger baseline performance, the model still experiences utility collapse after 12 concepts (Figure 10). Applying our proposed add-on regularizers yields improvements consistent with our previous benchmarks: all methods enhance retention performance over the baseline, with semantic-aware gradient-projection delivering the strongest results (Figure 11).

### A.2   ADDITIONAL ERASURE DOMAIN: CELEBRITY

To demonstrate that our findings generalize beyond style and object erasure, we present results for celebrity (identity-based) erasure. For the experimental setup, we select a sequence of 6 random celebrities to unlearn and an additional 6 for the held-out retention set. We employ the GIPHY celebrity classifier to measure unlearning accuracy (classifier error on unlearned celebrities) and retention accuracy (classifier accuracy on held-out celebrities). To further evaluate general retention performance, we generate 5,000 images using MS-COCO (Lin et al., 2014) prompts and report both FID and CLIP Score.

Consistent with our previous findings, ConAbl alone experiences severe utility degradation after just 6 celebrities, as shown in Figure 12. Furthermore, all proposed add-on regularizers improve retention capabilities without sacrificing unlearn-

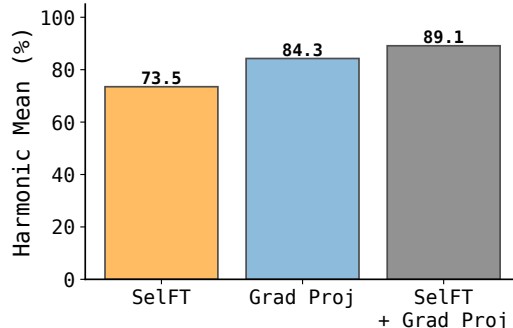

Figure 14: Gradient-projection is compatible with add-on regularizers, yielding additional gains when combining them. The y-axis shows the harmonic mean of UA, RA-I, and RA-C, using ConAbl for *celebrity* unlearning.

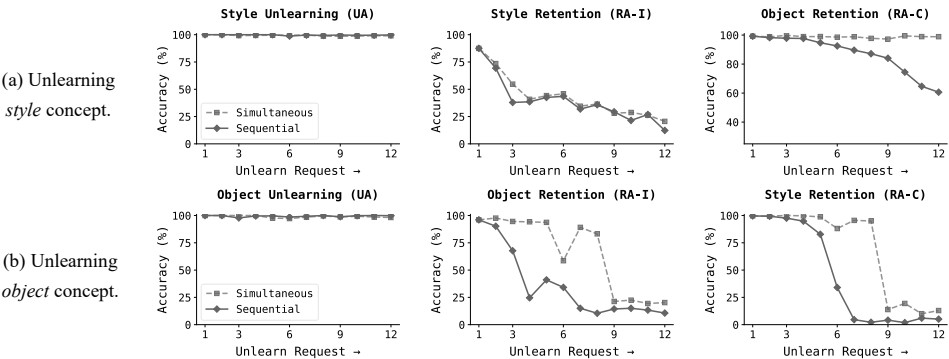

Figure 10: Performance of SculpMem (Li et al., 2025) under continual unlearning.

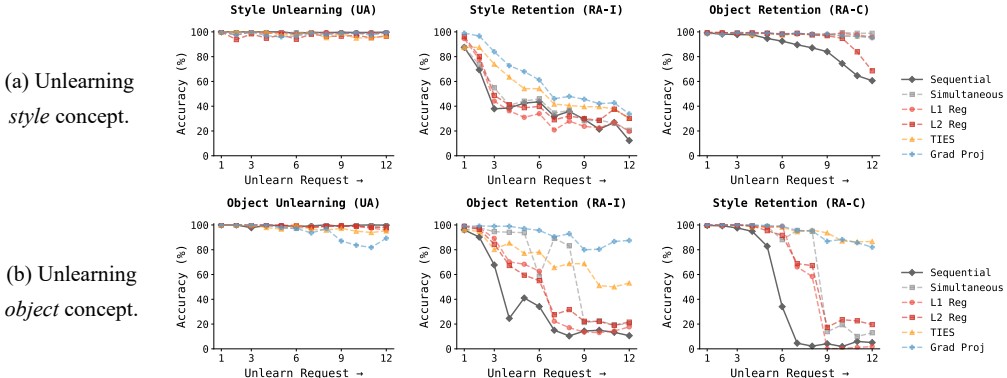

Figure 11: Continual unlearning with SculpMem using add-on regularizers.

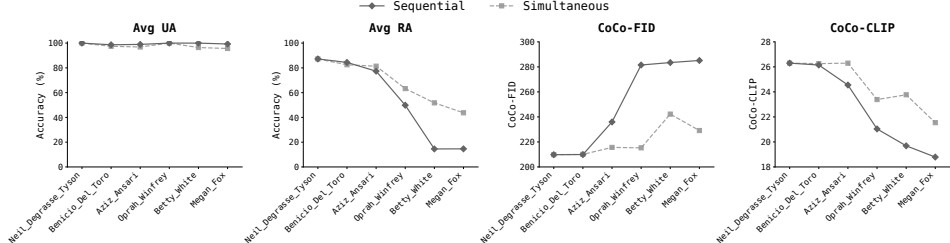

Figure 12: Performance of ConAbl (Kumari et al., 2023) under celebrity continual unlearning.

ing accuracy (Figure 13). The greatest performance gains come from combining our semantic-aware gradient-projection method with SelFT (Figure 14), demonstrating that add-on regularizers can be effectively combined for enhanced performance across different erasure settings.

## A.3 ADDITIONAL ARCHITECTURE: SDXL

Next, to demonstrate that our findings generalize to different architectures, we present results for SDXL using ESD (Gandikota et al., 2023b) for celebrity erasure. We adopt ESD rather than ConAbl or SculpMem because, to our knowledge, these methods do not provide official SDXL implementations. We continue with celebrity erasure instead of UnlearnCanvas (Zhang et al., 2024b), as the latter requires an SDXL checkpoint fine-tuned on its benchmark styles and objects, which is not publicly available.

As shown in Figure 15, ESD also experiences utility collapse in the celebrity erasure domain. As shown in Figure 16, gradient-projection outperforms L1, L2, and SelFT but is surprisingly outperformed by Model Merge. However, Figure 17 demonstrates that gradient projection can be combined with model merge for further performance gains.

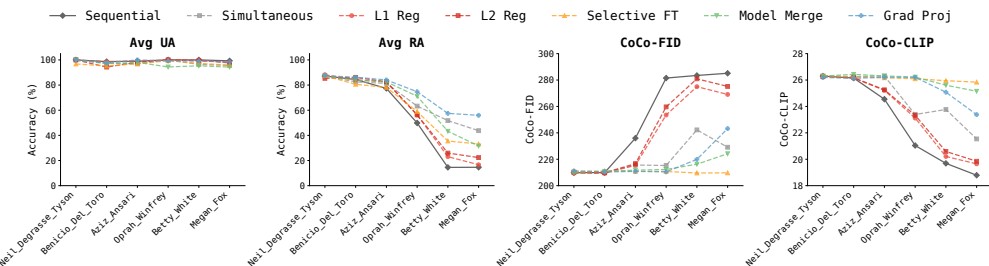

Figure 13: Celebrity continual unlearning with the ConAbl algorithm using add-on mechanisms.

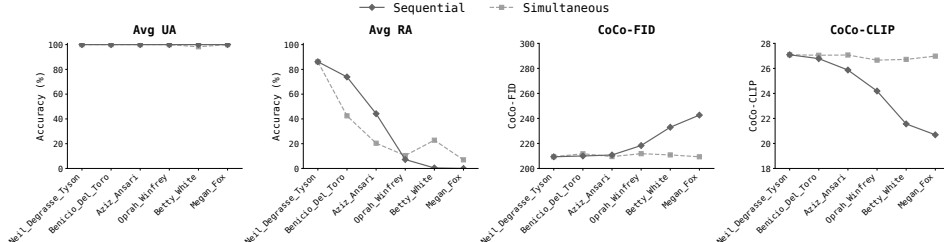

Figure 15: Performance of ESD (Gandikota et al., 2023b) under celebrity continual unlearning with SDXL.

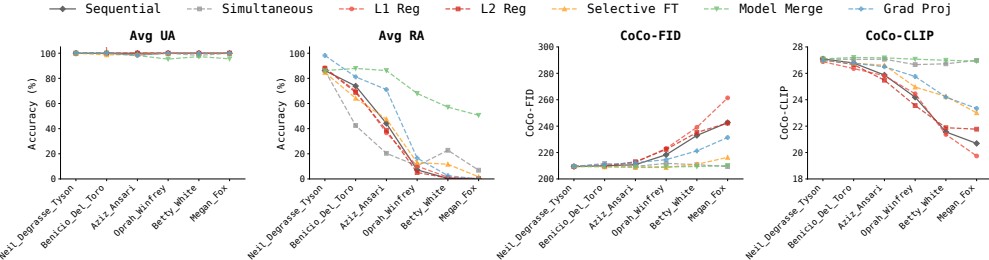

Figure 16: Celebrity continual unlearning with the ESD algorithm using add-on mechanisms on SDXL.

# B EXTENDED THEORETICAL SUPPORT

## B.1 FULL DERIVATION

Let $\theta^\star$ denote the model obtained from $\theta^\dagger$ after unlearning a concept (style or object), regardless of the unlearning method (*e.g.*, ConAbl, SculpMem, ESD), strategy (sequential or simultaneous), and add-on regularizers (*e.g.*, Gradient Projection, SelFT).

Let $\mathcal{C}$ be the set of all concepts learned by the diffusion model $\theta^\dagger$. Let $\mathcal{C}^f \subseteq \mathcal{C}$ be the set of concepts to be unlearned, and let $\mathcal{C}^r = \mathcal{C} \setminus \mathcal{C}^f$ be the remaining concepts to be retained.

Let $L^r(\theta; \mathcal{C}^r)$ denote the retention loss for preserving concepts in $\mathcal{C}^r$, instantiated in practice as the standard diffusion training objective.

We begin by applying a second-order Taylor expansion of the retention loss around $\theta^\dagger$ to approximate its value for any unlearned model $\theta^\star$.

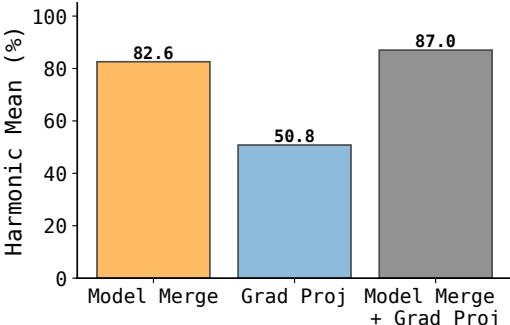

Figure 17: Gradient-projection is compatible with add-on regularizers, yielding additional gains when combining them. The y-axis shows the harmonic mean of UA, RA-I, and RA-C, using ESD (Gandikota et al., 2023a) with *SDXL* for *celebrity* unlearning.

$$L^r(\theta^\star; \mathcal{C}^r) = L^r(\theta^\dagger; \mathcal{C}^r) + \nabla L^r(\theta^\dagger; \mathcal{C}^r)^T(\theta^\star - \theta^\dagger) + \frac{1}{2}(\theta^\star - \theta^\dagger)^T H(\theta^\dagger)(\theta^\star - \theta^\dagger)$$

$$L^r(\theta^\star; \mathcal{C}^r) - L^r(\theta^\dagger; \mathcal{C}^r) = \nabla L^r(\theta^\dagger; \mathcal{C}^r)^T(\theta^\star - \theta^\dagger) + \frac{1}{2}(\theta^\star - \theta^\dagger)^T H(\theta^\dagger)(\theta^\star - \theta^\dagger)$$

The linear term, given by the inner product of the gradient and the parameter change, can be bounded via the Cauchy–Schwarz inequality:

$$|\langle \nabla L^r(\theta^\dagger; \mathcal{C}^r), (\theta^\star - \theta^\dagger)\rangle| \leq ||\nabla L^r(\theta^\dagger; \mathcal{C}^r)|| \cdot ||(\theta^\star - \theta^\dagger)||$$

The quadratic term can similarly be bounded through the repeated application of the Cauchy–Schwarz inequality, followed by the definition of the operator norm of the Hessian:

$$|\frac{1}{2}(\theta^\star - \theta^\dagger)^T H(\theta^\dagger)(\theta^\star - \theta^\dagger)| = \frac{1}{2}|\langle(\theta^\star - \theta^\dagger), H(\theta^\dagger)(\theta^\star - \theta^\dagger)\rangle|$$

$$\frac{1}{2}|\langle(\theta^\star - \theta^\dagger), H(\theta^\dagger)(\theta^\star - \theta^\dagger)\rangle| \leq \frac{1}{2}||\theta^\star - \theta^\dagger|| \cdot ||H(\theta^\dagger)(\theta^\star - \theta^\dagger)||$$

$$\frac{1}{2}||\theta^\star - \theta^\dagger|| \cdot ||H(\theta^\dagger)(\theta^\star - \theta^\dagger)|| \leq \frac{||H(\theta^\dagger)||}{2} \cdot ||\theta^\star - \theta^\dagger||^2$$

By substituting the bounds on the linear and quadratic terms, we obtain an overall bound on the change in retention loss between $\theta^\star$ and $\theta^\dagger$

$$|L^r(\theta^\star; \mathcal{C}^r) - L^r(\theta^\dagger; \mathcal{C}^r)| \leq ||\nabla L^r(\theta^\dagger; \mathcal{C}^r)|| \cdot ||(\theta^\star - \theta^\dagger)|| + \frac{||H(\theta^\dagger)||}{2} \cdot ||\theta^\star - \theta^\dagger||^2$$

This bound resembles a Lipschitz-type continuity condition:

$$|L^r(\theta^\star; \mathcal{C}^r) - L^r(\theta^\dagger; \mathcal{C}^r)| \leq L \cdot ||(\theta^\star - \theta^\dagger)|| + \frac{M}{2} \cdot ||\theta^\star - \theta^\dagger||^2$$

Thus, the change in retention loss grows proportionally with the parameter difference, with constants $L$ and $M$ bounding the contributions of the linear and quadratic terms, respectively.

The constants $L$ and $M$ are expected to be small when the pre-trained model is near a stationary point of the retention objective, and the local curvature of $L^r$ is low. In the next section, we provide an empirical approximation of $M$.

### B.2 EMPIRICAL SUPPORT

We empirically approximate the local gradient-Lipschitz constant of the retention objective via finite-difference gradient variation on held-out retention concepts. For each reference model $\theta$ (the base model and independently unlearned checkpoints), we sample perturbations

$$\delta = \epsilon||\theta||_2 u,$$

where $u$ is a random unit vector in the analyzed UNet subspace, and evaluate

$$\hat{M}(\epsilon) = \frac{||\nabla L^r(\theta + \delta; \mathcal{C}^r) - \nabla L^r(\theta; \mathcal{C}^r)||_2}{||\delta||_2}.$$

To reduce estimator noise, both gradients in each pair are computed on the same minibatch, diffusion noise realization, and timestep draw.

Across a logarithmic sweep of perturbation scales, we observe a consistent monotonic trend: in the smallest-perturbation regime, $\hat{M}(\epsilon)$ remains uniformly small across models, while larger perturbations yield larger $\hat{M}(\epsilon)$ and variance (Figure 18). This indicates that the retention objective is locally flat around the reference solution, but becomes increasingly nonlinear farther away.

These results support a local smoothness characterization: near the pretrained solution, the quadratic term in the Taylor expansion is weak; as perturbation magnitude increases, curvature effects become non-negligible and increasingly influence retention-loss change.

## C EXTENDED: CONCEPT ERASURE IS ALL-OR-NOTHING

We further analyze the interpolation behavior between the pre-trained model and its unlearned counterpart. Let $\theta^\alpha = (1 - \alpha)\theta^\dagger + \alpha \cdot \theta$ denote linear interpolation in parameter space. Across a wide range of $\alpha$, generated outputs for the target concept remain visually indistinguishable from the base model. However, once $\alpha$ crosses a concept-specific threshold, the target is abruptly suppressed Figure 19. This sharp transition indicates that erasure behaves in an all-or-nothing manner rather than degrading smoothly. This observation is further supported by classifier accuracy, which stays near 100% across most interpolation values and then sharply collapses to 0% at the transition point (Figure 20).

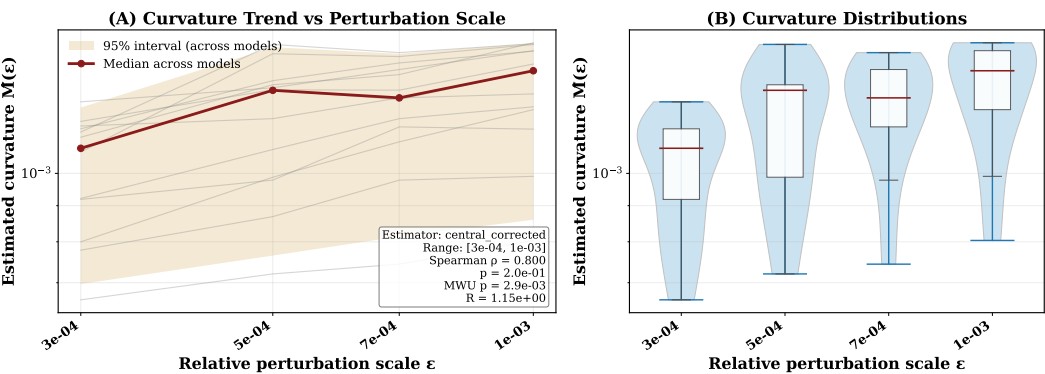

Figure 18: Estimated finite-difference retention curvature proxy under random UNet perturbations. For relative scale $\epsilon$, we compute $\hat{M}(\epsilon) = \|\nabla L^r(\theta + \delta; \mathcal{C}^r) - \nabla L^r(\theta; \mathcal{C}^r)\|_2 / \|\delta\|_2$ using matched minibatch, diffusion noise, and timesteps. (A) Per-model trends (thin lines) with median and 95% interval across models. (B) Distribution of $\hat{M}(\epsilon)$ at each scale. $\hat{M}(\epsilon)$ increases with perturbation scale, indicating that the retention objective is locally flat near the reference model and exhibits stronger nonlinear curvature farther away.

## D   EXTENDED: PARAMETER DRIFT IS INTRINSIC TO SEQUENTIAL UNLEARNING

In our original experimental design, sequential unlearning used a fixed number of iterations based on the default values recommended in the original unlearning method papers. For simultaneous unlearning, the relationship between training iterations and the number of concepts to unlearn was unknown. We therefore employed early stopping, evaluating unlearning accuracy every 100 iterations and terminating training once the model achieved 99% unlearning accuracy on a validation set. This approach also allowed simultaneous unlearning to serve as an initial upper-bound performance baseline for evaluating our sequential unlearning methods combined with the proposed add-ons.

However, to verify that the greater parameter drift in sequential versus simultaneous unlearning (Figure 4) is not caused by early stopping, we conduct a controlled experiment where early stopping is also applied to sequential unlearning. Specifically, we continue sequential unlearning until the cumulative number of optimization steps taken by simultaneous unlearning matches or exceeds those taken in the sequential setting (Table 1). Our revised heatmap (Figure 21) shows the same trend: parameter drift accumulates much faster in sequential than in simultaneous unlearning.

| Num Unlearned | Sequential Steps | Simultaneous Steps |
|:---:|:---:|:---:|
| 1 | 400 | 300 |
| 2 | 700 | 400 |
| 3 | 1200 | 800 |
| 4 | 1300 | 700 |
| 5 | 1900 | 1300 |
| 6 | 2100 | 2100 |

Table 1: Cumulative optimization steps for sequential and simultaneous unlearning when early stopping is applied to both settings. Each entry reports the total number of training steps required to erase the number of concepts indicated in the corresponding row. The cumulative step counts become equal by the sixth concept.

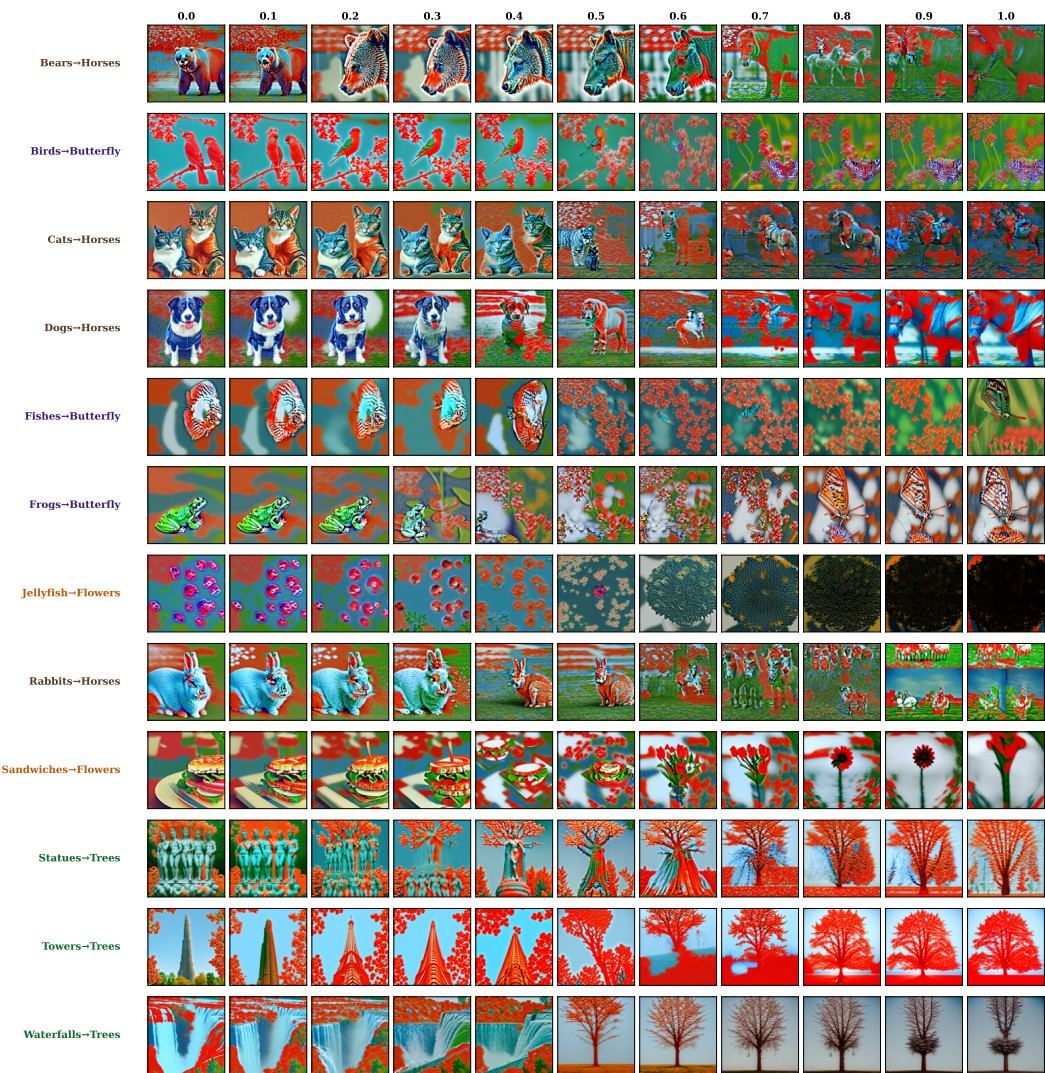

Figure 19: Qualitative interpolation between the pre-trained and unlearned models. For increasing interpolation coefficient $\alpha$, generated images remain visually unchanged until a critical threshold is reached, after which the target concept is abruptly suppressed.

# E    SIMULTANEOUS TRAINING COSTS

To compare the cumulative training costs of simultaneous and sequential unlearning, we perform style unlearning using ConAbl and utilize our best-performing add-on regularizer, semantic-aware gradient-projection. For fair comparison, we apply early stopping to both sequential and simultaneous unlearning, evaluating every 100 iterations and stopping once unlearning accuracy reaches 99%. As seen in Figure 22, sequential unlearning shows near-linear growth in training costs relative to the number of unlearning requests, while simultaneous unlearning exhibits superlinear growth. This is because simultaneous unlearning requires training from the base model at each unlearning request, thereby incurring repeated computation costs of re-unlearning previous requests. A comparison of unlearning and retention performance can be found in Figure 6.

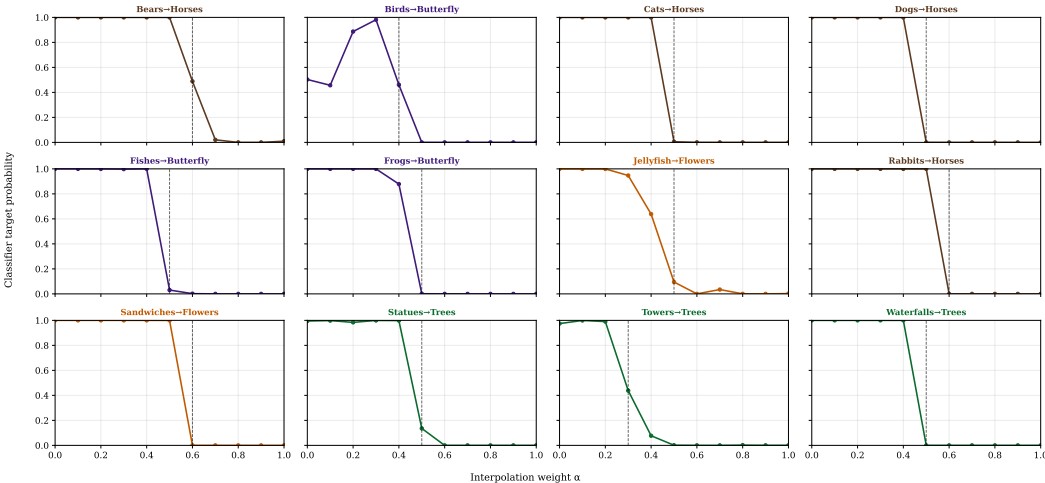

Figure 20: Quantitative evaluation of interpolation. Unlearning accuracy and retention metrics exhibit sharp transitions at concept-specific thresholds, supporting that erasure occurs as a sharp change rather than a gradual decline.

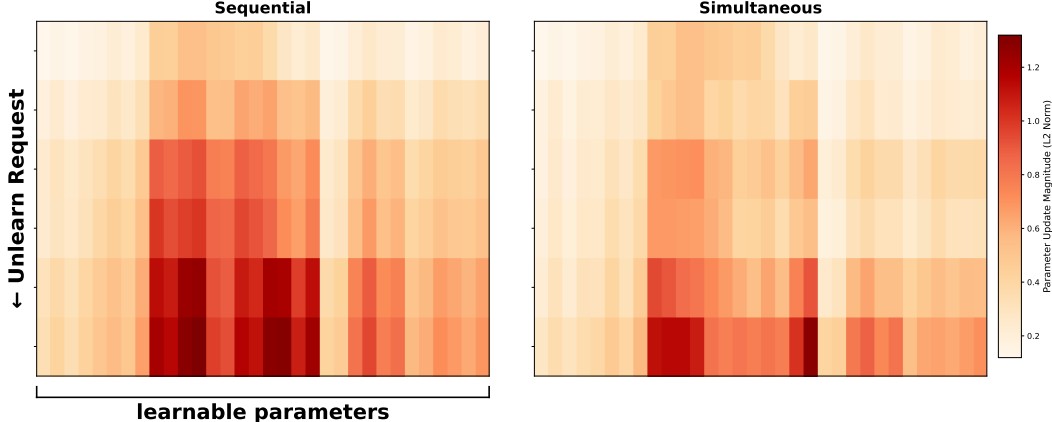

Figure 21: Parameter drift heatmap for ConAbl comparing sequential versus simultaneous unlearning when early stopping is applied to both strategies.

## F    DETAILED RELATED WORK

### F.1    FROM CONTINUAL LEARNING TO CONTINUAL UNLEARNING

Continual learning focuses on enabling models to acquire new knowledge incrementally without forgetting previously learned information—a phenomenon known as catastrophic forgetting (Mai et al., 2022; Wang et al., 2024; Lomonaco et al., 2022). Existing approaches to mitigate forgetting in continual learning can broadly be classified into four categories: (1) *regularization-based methods*, which incorporate explicit regularization terms to constrain parameter updates (Kirkpatrick et al., 2017; Zenke et al., 2017); (2) *replay-based methods*, which either store a limited set of previous examples in memory buffers (Mai et al., 2021; Shim et al., 2021) or employ generative models to synthesize replay samples (Shin et al., 2017); (3) *optimization-based methods*, which directly manipulate optimization procedures through techniques such as gradient projection (Chaudhry et al., 2018) or meta-learning (Javed & White, 2019); and (4) *architecture-based methods*, which introduce task-specific adaptive parameters to the model (Mallya et al., 2018).

Although continual unlearning fundamentally differs from continual learning, key concepts from continual learning remain valuable and adaptable (Heng & Soh, 2023). In this work, we leverage ideas

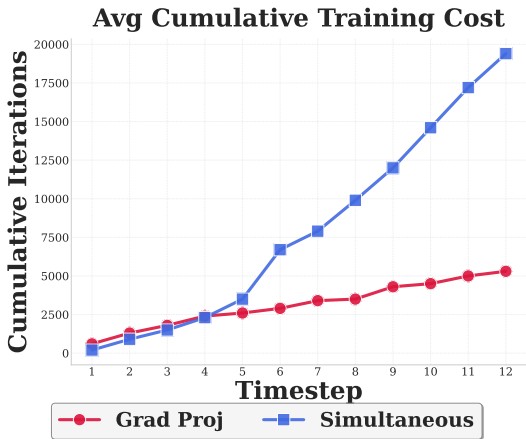

Figure 22: Cumulative training iterations for style unlearning with ConAbl, comparing sequential unlearning augmented with our best-performing add-on regularizer, semantic-aware gradient-projection, against simultaneous unlearning. Sequential unlearning exhibits near-linear growth in cumulative cost, whereas simultaneous unlearning incurs superlinear growth.

inspired by regularization-based methods from continual learning, introducing L1/L2 regularization baselines. Additionally, while selective parameter updates appear in both paradigms, continual learning methods update the least important parameters to preserve prior knowledge (Mazumder et al., 2021). In contrast, our proposed Selective Fine-Tuning (SelFT) approach identifies and updates the most significant parameters to facilitate effective unlearning.

By bridging insights from continual learning to continual unlearning, our research sets the stage for future investigations. We encourage subsequent studies to further integrate and refine continual learning strategies to address the nuanced challenges of continual unlearning effectively.

### F.2 SELECTIVE FINE-TUNING

Selecting the most important parameters within a model for a specific task has been extensively investigated for different purposes. To enhance time and memory efficiency, weight pruning methods commonly utilize gradient-based metrics to quantify parameter importance, enabling the removal of redundant parameters (Lee et al., 2018; Molchanov et al., 2016; Tanaka et al., 2020). A similar concept underlies model editing techniques, which aim to precisely locate and alter specific knowledge within a model by directly modifying relevant weights (Dai et al., 2021; Patil et al., 2023; De Cao et al., 2021). Recent work has extended these ideas to unlearning in diffusion models (Fan et al., 2023; Nguyen et al., 2024). Our findings demonstrate that incorporating selective fine-tuning into existing unlearning methodologies significantly enhances their performance in continual unlearning scenarios.

### F.3 SEMANTIC AWARENESS IN UNLEARNING

Most unlearning work emphasizes preserving model utility during concept removal. Beyond aggregate utility, it is equally important to identify which concepts are most susceptible to collateral degradation. Bui et al. (2025) investigate cross-concept effects and report that unlearning a concept disproportionately degrades semantically similar concepts. Complementarily, Bui et al. (2024) show that explicitly preserving closely related concepts yields larger overall utility retention. In contrast, we adopt a regularization perspective: we demonstrate that text-embedding similarity is a strong predictor of degradation and link this behavior to the cross-attention mechanism in diffusion models, where $W_K, W_V$ couple text directions with image latents.

### F.4 MODEL MERGING

Early research on model merging focused on averaging parameters of multiple models trained with varied hyperparameters on identical datasets to enhance generalization (Wortsman et al., 2022).

Concurrently, this strategy has been extended to multi-task learning, where models trained on diverse vision tasks have their weights averaged to achieve improved performance (Matena & Raffel, 2022; Ilharco et al., 2022). Since then, numerous advanced methods have emerged to refine the basic merging approach (fine-tuning followed by merging), including linearized fine-tuning (Ortiz-Jimenez et al., 2023), sparsifying update vectors (Davari & Belilovsky, 2024; Yu et al., 2024), and selectively merging subsets of weights (Yadav et al., 2023; Stoica et al., 2023).

Recent concurrent studies have also explored model merging techniques specifically tailored for unlearning in large language models (LLMs) (Kuo et al., 2025; Kadhe et al., 2024). However, to the best of our knowledge, this paper presents the **first** exploration of model merging for unlearning within the context of text-to-image generation.

