# OpenReview forum: "Continual Unlearning for Text-to-Image Diffusion Models: A Regularization Perspective"
_ICLR.cc/2026/Conference — ICLR 2026 Poster_

### Official Review · Reviewer_oE7J · 2025-10-26

**Soundness:** 3
**Presentation:** 3
**Contribution:** 2
**Rating:** 6
**Confidence:** 4

**Summary:**

This paper introduces the problem of continual unlearning for text-to-image diffusion models, where forgetting requests are handled sequentally. The paper shows that popular unlearning frameworks suffer from degraded performance in the continual setting. The paper shows that the problem is due to cumulative parameter drift, where the model weights progressively drift from the original weights. To address this, the paper proposes several methods: regularization, selective fine-tuning, model merging and gradient projection. Experiments show that these methods significantly improve the model's ability to retain performance in the continual unlearning setting.

**Strengths:**

- The continual unlearning setting is a practical and important problem in the field of text-to-image unlearning.
- While the root cause being cumulative parameter drift is unsurprising, the paper provides clear empirical evidence and analysis of the phenomenon.
- Though tested on only two baselines (see weaknesses), I find that the experiments and results are thorough and provide good evidence of the author's claims.
- Proposed methods like regularization and gradient projection are simple yet general, making them easy to integrate with existing unlearning pipelines.
- Overall, the paper is clear, concise and well-structured.

**Weaknesses:**

- In itself, the proposed methods are not novel (regularization, gradient projection etc. are certainly not new), but applied to the setting of continual unlearning.

- The experiments are conducted on two methods, ConAbl and EraseDiff. While these are representative, it is unclear how the findings of the proposed methods would generalize to other classes of unlearning algorithms.

- The authors claim simultaneous unlearning is costly, but their proposed model merging is also costly given independent copies have to be unlearned. Have the authors compared the computational costs and whether model merging is more efficient than simultaneous unlearning?

- On the Taylor expansion of the loss in Sec 5.2, the loss is *upper-bounded* by $||\theta^* - \theta^\dagger||$, thus even if the RHS grows, it does not guarantee the retention loss grows. Hence the conclusion that "loss grows proportionally (up to a constant) to $||\theta^* - \theta^\dagger||$" seems somewhat inaccurate.

**Questions:**

- Given the combined results in Fig 8, what end-to-end 'default' recipe do the authors recommend for sequential unlearning? A short set of takeaways for practitioners would help solidify the insights.
- Have the authors considered a Fisher-weighted regularization approach from continual learning like in [1]? This would lie between full L2 regularization and Selective fine-tuning.
- How sensitive is performance of gradient projection to the auxiliary set of semantically similar concepts? Could a poorly chosen set harm retention or unlearning?
- Have the authors investigated the effect of the order of unlearning requests? For example, does unlearning a broad concept (e.g. "photorealism") early in the sequence have a different impact than unlearning a more specific one (e.g. "Van Gogh")?
- How do results differ on concepts that can be referenced explicitly (like objects) versus indirectly (e.g., styles by prompting for artist names, or using synonyms)?

[1] Heng, Alvin, and Harold Soh. "Selective amnesia: A continual learning approach to forgetting in deep generative models." Advances in Neural Information Processing Systems 36 (2023): 17170-17194.

---

> ### Author Response · Authors · 2025-11-25
> **Rebuttal Part 1 of 4**
>
> We thank the reviewer for the constructive feedback and for recognizing that we propose a "practical and important problem". We also appreciate the acknowledgment that we provide "clear empirical evidence and analysis" and our presentation is "clear, concise, and well-structured." We address each concern raised by the reviewer below.
>
> ## (W1) Method Novelty
> Thank you for the comment. We want to reiterate that the primary goal of this paper is not to introduce entirely new methods, but to systematically study continual unlearning for text-to-image diffusion models and to understand why utility degradation occurs in this setting. Our findings show that the core problem lies in parameter drift and semantic confusion (Section 5.2, Section 7).
>
> This insight led us to investigate existing approaches that can potentially address these issues. While we recognized that choosing these existing methods may hurt our novelty, we humbly believe this is the most appropriate first step: **grounding our work in existing literature**. Specifically, we chose a diverse set of methods (L1, L2, Selective Fine-Tuning, Model Merge), each emphasizing different aspects of preventing drift.
>
> Moreover, we find that concepts closer in text embedding space to the unlearning target are more susceptible to utility degradation, which in turn motivates our semantic-aware gradient projection approach. While gradient projection alone is widely used in machine learning, in our humble opinion, its targeted application for preventing drift paired with semantically related concept sets is novel.
>
> ## (W2) Finding Generalizability
> Thank you for pointing out this concern about generalizability. To evaluate whether our findings extend beyond ConAbl and EraseDiff, we have conducted additional experiments on **Sculpting Memory** (*ICCV 2025, suggested by Reviewer 6PKG*), a recent SOTA multi-concept erasure method. We benchmark Sculpting Memory on the UnlearnCanvas benchmark under our continual unlearning setting, with and without all of our proposed add-ons.
>
> As expected for a SOTA method, Sculpting Memory is indeed more robust than ConAbl and EraseDiff in the continual setting. However, we note that the original Sculpting Memory paper evaluates at most 5 concepts, whereas our setting extends to 12 sequentially unlearned concepts. Under this longer horizon, we observe the same failure mode: utility preservation starts strong but degrades substantially; by the 12th concept, both style and object in-domain retention drop below 15% (Appendix Figure 12).
>
> Consistent with our earlier findings, adding our semantic-aware gradient projection to Sculpting Memory yields the best in-domain retention among all add-ons. The gains for object unlearning are particularly pronounced: by the 12th concept, our projection achieves 87.50% in-domain retention and 82.08% cross-domain retention, compared to 10.62% and 5.08% respectively without our add-on (Appendix Figure 13).
>
> These new experiments continue to support two key claims:
>
> - Cumulative drift and semantic confusion are fundamental issues that persist across distinct families of unlearning algorithms, not just ConAbl and EraseDiff.
>
> - Our proposed methods are base-agnostic, plug-and-play modules that consistently improve in-domain and cross-domain retention when attached to diverse unlearning backbones.
>
> We will update the paper to include these Sculpting Memory results and to more clearly emphasize that our contributions are designed to be complementary to existing and future unlearning methods, rather than tied to a specific algorithmic class.

---

> ### Author Response · Authors · 2025-11-25
> **Rebuttal Part 2 of 4**
>
> ## (W3) Cost Analysis
>
> To provide a holistic cost analysis of baselines and our add-ons, we present both memory usage and runtime analysis in the table below.
>
> | Domain | Iterations | Metric | Base | L1 | L2 | SelFT | Merge | Projection |
> |--------|------------|--------|------|-----|-----|-------|-------|------------|
> | Style | 1000*N | Runtime (min) | 14.5 (N) | 0.2(N) | 0.2(N) | 2.6(N) | 0.1(N) | 1.38(N)  |
> | | | Storage (MB) | 73.1(1) | 0 | 0 | 0 | 73.1(N-2) | 0 |
> | Object | 2000*N | Runtime (min) | 29.6(N) | 0.2(N) | 0.2(N) | 2.6(N) | 0.1(N) | 1.38(N) |
> | | | Storage (MB) | 73.1(1) | 0 | 0 | 0 | 73.1(N-2) | 0 |
>
> In the provided table, we present runtime and storage analysis for Concept Ablation across both style and object erasure domains. Iterations denote the number of training iterations, and Base shows the number of minutes and megabytes incurred by the base unlearning method when used sequentially. The notation (N) indicates that cost grows linearly with the number of erasure requests, while (1) denotes constant cost. The columns L1, L2, Selective Fine-Tuning (SelFT), Merge, and Projection show additional costs relative to Base.
>
> All methods incur **small additional runtime costs**. Projection takes 1.38 minutes due to the LLM API call made to generate auxiliary concept captions for forming the text embedding subspace.
>
> All methods except Model Merge incur **no additional storage costs**. Model Merge requires access to all (N-1) previously trained models at the N-th request, therefore incurring an additional (N-2) storage cost relative to Base. However, the merging operation is very efficient and runs entirely on CPU.
>
> It is important to note that compared to simultaneous unlearning, sequential methods become significantly **scalable** as the number of erasure concepts increases. Simultaneous unlearning requires re-unlearning **all the concepts** from the base model at each request, incurring repeated computation that grows rapidly with scale. As shown in Figure x in the Appendix, our sequential unlearning maintains linear growth in training iterations, while simultaneous unlearning exhibits exponential growth. This exponential curve demonstrates that the number of required training iterations rapidly increases with each additional unlearning request, making sequential approaches substantially more efficient for multiple erasure concepts.
>
> ## (W4) Taylor Expansion Clarification
> Thank you for this careful observation. We would like to respectfully clarify a misunderstanding regarding the implications of our theoretical result.
>
> Firstly, we want to highlight that LHS represents the change in retention loss between the base model and the fine-tuned version derived from it. Retention loss represents the ability to generate images for unrelated concepts. Thus, in the continual unlearning setting, we want this change to be as small as possible, such that the fine-tuned model’s ability to generate images for unrelated concepts is similar to the original model.
>
> In our Taylor-based argument, the retention loss difference between the base model and its fine-tuned version is **upper-bounded** by a constant times $\|\theta^* - \theta^\dagger\|$; this bound does not imply that the loss *must* increase whenever the parameter distance increases.
>
> Our intended claim is more modest:
>
> - The change in retention loss (LHS) is **controlled in the worst case** by the amount of parameter drift (RHS).
> - Therefore, by reducing $\|\theta^* - \theta^\dagger\|$, we can provably shrink the **maximum possible** degradation in retention loss.
>
> This provides theoretical justification for our proposed approaches, which aim to limit parameter drift and thereby constrain the potential magnitude of change in retention loss.
>
> In the revision, we will clarify that our theoretical result gives an **upper bound** that motivates drift-minimizing methods, rather than a strict monotonic relationship.
>
> ## (Q1) Default Recipe for Sequential Unlearning
>
> We appreciate the reviewer’s suggestion to distil our findings into a practical “default recipe” for sequential unlearning. Based on the combined results in Fig. 8 (and the additional experiments added in the rebuttal), our main takeaways for practitioners are:
>
> 1. **Start from a strong base unlearning method.**
>    Use any SOTA  unlearning method that works well for single / multiple concepts. Our add-ons are designed to be **base-agnostic** and plug in on top of these methods.
> 2. **Default recipe for continual unlearning:**
>    - **Combine SelFT with Gradient Projection** to explicitly control semantic confusion and mitigate cumulative parameter drift.
> 3. **Optionally add Model Merging**
>  We recommend also applying **Model Merge** between the unlearned model and its pre-unlearning checkpoint.
>
> We will add a short “practitioner takeaway” subsection in the revision to clearly summarize these guidelines.

---

> ### Author Response · Authors · 2025-11-25
> **Rebuttal Part 3 of 4**
>
> ## (Q2) Fisher Regression
> We thank the reviewer for this thoughtful suggestion. We had independently explored a similar Fisher-weighted regularization approach. Our initial experiments showed that standard L1 regularization outperformed L2, which motivated us to reweight the L1 penalty using the inverse Fisher Information Matrix. The goal was to discourage updates to parameters that were unlikely to be important for unlearning and to promote more targeted sparsity. In practice, however, this approach underperformed relative to standard L1 regularization in our continual unlearning setting.
>
> We hypothesize this occurs because unlearning gradients provide less informative signals for estimating Fisher information compared to standard learning gradients. During unlearning, the model receives signals to suppress specific concepts, which may not accurately capture which parameters are most critical for retaining other capabilities. In contrast, learning gradients directly indicate parameter importance for the target task, making Fisher information estimates more reliable in traditional continual learning settings. This fundamental difference in gradient informativeness may explain why Fisher-based approaches that prove effective for continual learning show limited benefits for continual unlearning.  However, we believe exploring alternative parameter importance estimation methods specifically designed for unlearning represents a promising direction for future research.
>
> **Fisher-weighted Regularization (Inverse EWC [1])**
>
> | Metric | 2 | 4 | 6 | 8 | 10 |
> |--------|---------|---------|---------|---------|---------|
> | UA | 93.75% | 98.12% | 97.50% | 99.06% | 99.75% |
> | RA-I | 71.46% | 25.21% | 20.62% | 14.38% | 11.46% |
> | RA-C | 98.21% | 99.69% | 99.17% | 97.50% | 95.91% |
>
> **L1**
>
> | Metric | 2 | 4 | 6 | 8 | 10 |
> |--------|---------|---------|---------|---------|---------|
> | UA | 95.00% | 94.38% | 93.33% | 92.81% | 94.50% |
> | RA-I | 74.17% | 30.62% | 26.25% | 19.58% | 14.79% |
> | RA-C | 98.39% | 98.91% | 96.81% | 92.75% | 90.91% |
>
> [1]  Loke, B. S. Y., Quadri, F., Vivanco, G., Casagrande, M., & Fenollosa, S. (2025). Overcoming catastrophic forgetting in neural networks. arXiv preprint arXiv:2507.10485.
>
> ## (Q4) Unlearn Request Order
>
> We thank the reviewer for this insightful question. To examine order effects, we conducted additional experiments on both style and object domains using three alternative unlearning sequences: (i) the original sequence reversed, and (ii–iii) two independently shuffled random sequences. For each metric, we report the mean and standard deviation across these sequences and the original to quantify sensitivity to order.
>
> Our results show that the sequence order induces **moderate variance** in retention performance for the first few concepts, but this variance diminishes rapidly once around eight concepts have been unlearned. We interpret this as evidence that some concepts are indeed more harmful to model utility than others, so unlearning a particularly “strong” or broad concept early can accelerate degradation. However, as parameter drift accumulates over many unlearning steps, overall utility eventually collapses and the retention curves across different orders largely converge.
>
> Importantly, although different orders yield different absolute performance levels, our proposed add-ons consistently improve retention over all base unlearning methods for all sequences we tested, indicating that they are **robust** to unlearning sequences.
>
> **Base Sequential**
>
> | Metric | 1 | 3 | 5 | 8 | 12 |
> |--------|---------|---------|---------|---------|---------|
> | UA | 100.00% ± 0.00% | 99.58% ± 0.76% | 98.50% ± 0.60% | 98.80% ± 1.08% | 99.27% ± 0.97% |
> | RA-I | 93.18% ± 3.77% | 64.95% ± 12.38% | 46.61% ± 18.90% | 15.99% ± 3.86% | 11.69% ± 3.17% |
> | RA-C | 98.75% ± 0.94% | 86.33% ± 5.29% | 69.07% ± 9.96% | 15.03% ± 9.11% | 4.35% ± 1.96% |
>
>
> **Gradient Projection**
>
> | Metric | 1 | 3 | 5 | 8 | 12 |
> |--------|---------|---------|---------|---------|---------|
> | UA | 99.58% ± 0.76% | 99.03% ± 1.94% | 96.75% ± 5.37% | 97.40% ± 1.39% | 97.72% ± 1.66% |
> | RA-I | 99.01% ± 0.29% | 98.33% ± 0.69% | 96.51% ± 0.94% | 81.41% ± 5.72% | 61.77% ± 7.83% |
> | RA-C | 99.59% ± 0.68% | 97.39% ± 2.83% | 95.58% ± 2.71% | 83.67% ± 9.72% | 50.98% ± 8.86% |

---

> ### Author Response · Authors · 2025-11-25
> **Rebuttal Part 4 of 4**
>
> ## Q(3) Hyperparameter Sensitivity
>
> We thank the reviewer for raising this concern. To provide a more comprehensive analysis of each proposed add-on, we present a hyperparameter sensitivity study in the table below using Sculpting Memory on style erasure. Starting from our chosen hyperparameter values, we evaluate performance when each value is scaled by factors of 0.5 and 2.0. The table reports the mean performance along with standard deviation to quantify sensitivity to hyperparameter selection.
>
>
> We observe that our add-ons are generally robust to hyperparameter changes. For example, for the gradient projection and model merge methods, most standard deviations are lower than 3%.  Given this stability, we use a **fixed** set of L1/2 ($\lambda$=50), SelFT(top-k=5%), Gradient Projection(auxiliary concept number=200) for all experiments,  with sole exception of the model merging factor, which benefits from minor tuning.
>
> Notes that in the tables below, UA is the unlearn accuracy, RA-I is the in-domain retention accuracy, and RA-C is the cross-domain retention accuracy.
>
> **Model Merge**
> | Metric | Abstr(1) | Cartoon(3) | Ukiyoe(5) | Picasso(8) | Vibrant(12) |
> |--------|----------|------------|-----------|------------|-------------|
> | UA | 100.00% ± 0.00% | 100.00% ± 0.00% | 98.75% ± 1.77% | 97.66% ± 3.32% | 98.40% ± 2.28% |
> | RA-I | 85.35% ± 0.98% | 72.40% ± 2.22% | 51.57% ± 3.69% | 39.58% ± 1.47% | 28.99% ± 1.72% |
> | RA-C | 99.23% ± 0.25% | 99.17% ± 0.00% | 98.16% ± 0.85% | 98.12% ± 0.00% | 95.67% ± 0.37% |
>
>
> **Grad Proj**
> | Metric | Abstr(1) | Cartoon(3) | Ukiyoe(5) | Picasso(8) | Vibrant(12) |
> |--------|----------|------------|-----------|------------|-------------|
> | UA | 100.00% ± 0.00% | 100.00% ± 0.00% | 98.00% ± 2.83% | 98.91% ± 1.54% | 99.17% ± 0.00% |
> | RA-I | 98.44% ± 0.52% | 80.00% ± 5.60% | 67.92% ± 1.21% | 53.65% ± 3.11% | 37.44% ± 2.22% |
> | RA-C | 98.46% ± 0.27% | 98.67% ± 0.94% | 98.68% ± 0.42% | 97.06% ± 1.33% | 95.37% ± 0.22% |
>
>
> **SelFT**
> | Metric | Abstr(1) | Cartoon(3) | Ukiyoe(5) | Picasso(8) | Vibrant(12) |
> |--------|----------|------------|-----------|------------|-------------|
> | UA | 100.00% ± 0.00% | 99.59% ± 0.59% | 99.50% ± 0.71% | 99.54% ± 0.22% | 99.90% ± 0.15% |
> | RA-I | 92.29% ± 6.79% | 67.50% ± 5.66% | 40.42% ± 5.92% | 32.29% ± 2.05% | 14.59% ± 2.88% |
> | RA-C | 99.23% ± 0.00% | 98.61% ± 0.56% | 97.20% ± 3.54% | 98.50% ± 1.93% | 96.79% ± 3.07% |
>
>
> **L1**
> | Metric | Abstr(1) | Cartoon(3) | Ukiyoe(5) | Picasso(8) | Vibrant(12) |
> |--------|----------|------------|-----------|------------|-------------|
> | UA | 92.50% ± 10.61% | 95.00% ± 7.07% | 96.50% ± 4.24% | 97.82% ± 2.65% | 98.86% ± 0.16% |
> | RA-I | 95.83% ± 1.77% | 53.75% ± 13.56% | 33.96% ± 4.13% | 27.19% ± 1.03% | 20.21% ± 0.88% |
> | RA-C | 98.85% ± 0.01% | 99.34% ± 0.47% | 98.97% ± 0.00% | 98.50% ± 0.54% | 97.71% ± 2.66% |
>
> **L2**
>
> | Metric | Abstr(1) | Cartoon(3) | Ukiyoe(5) | Picasso(8) | Vibrant(12) |
> |--------|----------|------------|-----------|------------|-------------|
> | UA | 97.50% ± 3.54% | 95.83% ± 3.54% | 97.00% ± 0.71% | 97.03% ± 0.66% | 97.19% ± 0.59% |
> | RA-I | 96.98% ± 1.06% | 52.92% ± 5.89% | 37.19% ± 2.21% | 32.92% ± 1.18% | 28.34% ± 2.84% |
> | RA-C | 99.52% ± 0.14% | 98.83% ± 0.00% | 98.82% ± 0.00% | 98.00% ± 1.06% | 80.94% ± 1.68% |
>
> ## (Q5) Relationship Between Concepts with Many Synonyms
>
> We thank the reviewer for this thoughtful question. In our current setup, we explicitly separate concepts that can be referenced directly (objects) from those that are more often referenced indirectly via artist names or stylistic cues (styles). As shown in Figure 6, we find that object and style concepts exhibit **similar qualitative trends**: erasure remains effective while retention gradually degrades as continual unlearning progresses. At the same time, there are quantitative differences in how quickly and how severely retention deteriorates across the two domains.
>
> The more fine-grained distinction the reviewer raises between concepts that can be referenced *explicitly* versus those that admit many *indirect* references or synonyms (e.g., “Renaissance art” vs. “Humanist art”) is not explicitly controlled in our current benchmark.
>
> To the best of our knowledge, this dimension has not been systematically studied in prior unlearning work either. We view it as an interesting and complementary research direction that focuses on the inherent difficulty of fully unlearning a **single** concept under diverse linguistic realizations, whereas our paper is primarily concerned with whether an algorithm can **continually** unlearn *multiple* concepts over time without catastrophic utility loss.

---

### Official Review · Reviewer_xbNs · 2025-10-28

**Soundness:** 3
**Presentation:** 3
**Contribution:** 3
**Rating:** 6
**Confidence:** 3

**Summary:**

This work conducts the first systematic investigation of continual unlearning in text-to-image diffusion models, revealing that existing unlearning methods suffer from severe utility collapse due to cumulative parameter drift. To address this, the authors propose a set of regularization-based strategies that mitigate drift while remaining compatible with existing methods. They further introduce a semantic-aware gradient projection technique that constrains parameter updates to directions orthogonal to the target concept’s subspace, preserving related knowledge. Overall, these methods substantially improve continual unlearning stability and establish strong baselines for safe, accountable generative AI.

**Strengths:**

1. This paper presents an interesting and valuable study on continual unlearning in text-to-image diffusion models.
2. This paper is very well presented.
3. The paper conducts a detailed analysis of the challenges faced by continual unlearning in text-to-image diffusion models through a series of experiments.

**Weaknesses:**

The author emphasizes that this paper does not propose new algorithms but focuses on the analysis of continual unlearning. I have the following questions about this paper:

1. Compared to regular continual learning, what are the additional challenges of continual unlearning? Parameter drift and conceptual confusion have been extensively studied in continual learning. Results in figure 3 separate the unleaning target from the retention target, but can also be interpreted as follows: as the number of requests increases, the model forgets the required targets, leading to indiscriminate unleaning of all concepts.
2. Experimental findings indicate that object retention and style retention exhibit distinct patterns of forgetting, though further analysis of this phenomenon is lacking.
3. In the experiment shown in Figure 4, for sequential learning, is the sum of the update iterations for multiple requests the same as the update iterations for simultaneous learning?
4. What insights does this paper offer for future research on continual unlearning in text-to-image diffusion models. Given that the methods combined in this paper have long been applied in continual learning, does this imply that regularization techniques and model merging approaches designed to address continual learning issues can effectively tackle continual unlearning in text-to-image diffusion models?

**Questions:**

Minor concerns:

- The colors in Figure 3 are too similar, resulting in poor readability.

---

> ### Author Response · Authors · 2025-11-25
> **Rebuttal Part 1 of 2**
>
> We thank the reviewer for their insightful questions and for recognizing our work as "interesting and valuable" with "detailed analysis" that is "very well presented." Below, we address each question raised by the reviewer.
>
> ## (W1) Continual Learning vs Continual Unlearning
> We thank the reviewer for this critical question about the relationship between continual learning and continual unlearning. As discussed in Section 2, the two settings are indeed closely related: both aim to update an existing model while balancing the competing objective of preserving existing capabilities. However, the key distinction is that in continual unlearning, both the concept to be removed and the concepts to be retained are already encoded within the model. In contrast, continual learning involves learning new concepts, which may introduce different **interference dynamics**.
>
> We fully agree with the reviewer’s interpretation of Figure 3. One of the contributions of our work is identifying that one of the root causes of forgetting in continual learning, parameter drift, is also responsible for failures in continual unlearning. Thus, as mentioned in Section 2 and Appendix E, we believe principles from continual learning remain highly relevant to continual unlearning.
>
> Consequently, rather than immediately proposing a novel unlearning method, we choose a more appropriate first step: grounding on existing literature. We ask: *to what extent can techniques from continual learning mitigate the collapse we observe?* To answer this, we systematically evaluate a diverse set of add-ons (L1, L2, SelFT, Model Merge, Gradient Projection), each targeting a different aspect of drift and interference.
>
> Beyond evaluating existing methods, we proposed a Gradient Projection method tailored specifically for text-to-image unlearning. This method is theoretically grounded and based on our observation that concepts semantically close to the unlearning target in text embedding space are most susceptible to degradation.
>
> Thus, we consider our work as a **bridge** to connect continual learning to continual unlearning, aiming to set a solid foundation for future research to refine these strategies.
>
> ## (W2) Analysis on Style vs Object Unlearning
> The reviewer is correct that object and style unlearning exhibit distinct patterns of forgetting. Empirically, we observe that styles are easier to unlearn but more difficult to retain. As shown in Figure 3, when unlearning styles, style retention degrades smoothly and consistently throughout the unlearning sequence. In contrast, when unlearning objects, object retention remains high until reaching a critical threshold, at which point it declines abruptly.
>
> We hypothesize that this asymmetry arises from how diffusion models internally represent different types of information. Prior work has shown that diffusion models generate images in a *coarse-to-fine* manner, where global shape and structure are established in early timesteps and are driven by low-frequency signals, while finer texture details are refined later in the process [2, 3]. Together with evidence that modern generative models exhibit a stronger *shape bias* than *texture bias* [1], this suggests that object-level structure is encoded more robustly and more centrally in the model’s representation space than style-related texture cues. As a result, object information is more resistant to forgetting, whereas styles—being more associated with finer, texture-like details—are more easily erased but also more fragile to retain.
>
> We also conducted empirical analysis to test this hypothesis that styles are easier to unlearn. We measured the minimum number of training iterations required to erase 10, 11, and 12 styles versus objects, holding all hyperparameters constant (learning rate, batch size, etc.). Every 100 iterations, we evaluated whether unlearning accuracy had reached 100%. We found that styles require significantly fewer iterations to unlearn the same number of concepts. While this empirical test is not conclusive, we believe it provides insights into the differing difficulty of removing and retaining styles versus objects.
>
>
> | Num Unlearned | Style Iterations | Object Iterations |
> |---------------|------------------|-------------------|
> | 10            | 2600             | 4700              |
> | 11            | 2600             | 3800              |
> | 12            | 2200             | 4500              |
>
> [1] Jaini, P., Clark, K., & Geirhos, R. (2023). Intriguing properties of generative classifiers. arXiv preprint arXiv:2309.16779.
>
> [2] Saharia, C., Ho, J., Chan, W., Salimans, T., Fleet, D. J., & Norouzi, M. (2022). Image super-resolution via iterative refinement. IEEE transactions on pattern analysis and machine intelligence, 45(4), 4713-4726.
>
> [3] Huang, X., Salaun, C., Vasconcelos, C., Theobalt, C., Oztireli, C., & Singh, G. (2024, July). Blue noise for diffusion models. In ACM SIGGRAPH 2024 conference papers (pp. 1-11).

---

> ### Author Response · Authors · 2025-11-25
> **Rebuttal Part 2 of 2**
>
> ## (W3) Sum of iterations for sequential vs simultaneous
> We appreciate the reviewer raising this question. In our experimental setup, sequential unlearning used a fixed number of iterations based on the default values recommended in the original unlearning method papers. For simultaneous unlearning, the relationship between training iterations and the number of concepts to unlearn was unknown. We therefore employed early stopping, evaluating unlearning accuracy every 100 iterations and terminating training once the model achieved 99% unlearning accuracy on a validation set. This approach also allowed simultaneous unlearning to serve as an upper-bound performance baseline for evaluating our sequential unlearning methods combined with the proposed add-ons.
>
> To study the impact of iterations, we repeated the experiment with a controlled setup where early stopping is applied consistently to both sequential and simultaneous conditions. Our revised heatmap in Appendix Figure 26 confirms the original finding: sequential unlearning continues to accumulate significantly more parameter drift compared to simultaneous unlearning, even when early stopping is accounted for. Interestingly, this parameter drift phenomenon is independent of the number of training iterations. As the table below shows, by the sixth concept, both sequential and simultaneous utilize the same number of accumulated training iterations, yet sequential accumulates more parameter drift. The heatmap demonstrates that both the number of parameters significantly affected and the degree to which they are impacted are greater for sequential unlearning.
>
> | Num. Unlearned | Sequential Iterations | Simultaneous Iterations |
> |---------------|-----------------------|-------------------------|
> | 1             | 400                   | 300                     |
> | 2             | 700                   | 400                     |
> | 3             | 1200                  | 800                     |
> | 4             | 1300                  | 700                     |
> | 5             | 1900                  | 1300                    |
> | 6             | 2100                  | 2100                    |
>
> ## (W4) Future Research Insight
> Regarding guidance for future research on continual unlearning, we believe continual learning techniques provide a solid foundation that should continue to be explored. However, our findings highlight two critical directions for advancing the field.
>
> First, model merging shows particularly strong promise for continual unlearning scenarios. In our experiments, we were surprised to find that model merging performs remarkably well compared to sequential fine-tuning approaches. We hypothesize this success stems from each merge operation anchoring itself to the original base model rather than building on sequentially modified versions, which provides implicit regularization against parameter drift. This observation suggests that future work on model merging strategies specifically designed for unlearning could yield significant improvements.
>
> Second, and perhaps more importantly, our analysis reveals that semantic awareness appears even more critical for unlearning than for traditional continual learning tasks. Unlike adding new capabilities, removing existing knowledge creates interference patterns that are heavily influenced by semantic proximity in the learned representation space. Methods should be developed with explicit awareness that concepts closer in text embedding space are more likely to experience collateral degradation during unlearning. We believe combining model merging approaches with semantic-aware mechanisms represents a particularly promising research direction.
>
> ## (Q1) Figures
> Thank you for the feedback on Figure 3. We have updated the figure in the latest revision and improved the color scheme to make the distinctions clearer.

---

> > ### Comment · Reviewer_xbNs · 2025-11-26
> > **Response to Rebuttal**
> >
> > Thank you for the authors' in-depth analysis and experimental details, which addresses my concerns.

---

> > > ### Author Response · Authors · 2025-11-26
> > >
> > > We are grateful we were able to address the reviewer's concerns. We humbly ask whether there are any additional clarifications we can provide that might lead the reviewer to consider raising their score.

---

### Official Review · Reviewer_6PKG · 2025-10-30

**Soundness:** 3
**Presentation:** 3
**Contribution:** 3
**Rating:** 6
**Confidence:** 4

**Summary:**

The paper introduces continual unlearning for text-to-image diffusion models where removal requests arrive sequentially. It shows that popular methods degrade quickly in this setting due to cumulative parameter drift away from the pretrained weights, then proposes simple add-on remedies such as L1 or L2 update penalties, selective fine-tuning, model merging, and a semantic-aware gradient projection on cross-attention projections to protect nearby concepts. Experiments on an UNLEARNCANVAS-based benchmark report strong improvements in retention while maintaining unlearning effectiveness.

**Strengths:**

1. Clear problem definition of continual unlearning with precise requirements for erasing targets, preserving prior removals, and retaining unrelated capabilities, plus explicit metrics for unlearning accuracy and retention accuracy split into in-domain and cross-domain.

2.  Practical plug-and-play remedies that integrate with existing unlearning methods, including L1 or L2 update penalties, selective fine-tuning, and model merging via TIES, which reduce drift and improve retention.

3. The gradient projection idea operates on cross-attention projections to protect nearby concepts and combines well with other regularizers for further improvements.

**Weaknesses:**

1. Sensitivity to choices such as the strength of L1 or L2 penalties, the top k percent for selective updates, and the number and selection rule for auxiliary concepts in gradient projection is not fully characterized.

2. Limited cost analysis for independent unlearning plus merging and for importance computation in selective tuning.

3. The paper does not discuss several closely related recent works that address multi-concept and efficient forgetting, such as Sculpting Memory: Multi-Concept Forgetting in Diffusion Models via Dynamic Mask and Concept-Aware Optimization (ICCV 2025) and ConceptPrune: Concept Editing in Diffusion Models via Skilled Neuron Pruning (ICLR 2025). These studies provide complementary perspectives on dynamic masking and pruning-based forgetting, and should be compared for completeness.

**Questions:**

1. How sensitive are the results to the regularization coefficient and top k selection used in selective fine-tuning and merging?

2. How are auxiliary concepts selected for gradient projection, and how many are required for stable performance?

3. The evaluation primarily relies on EraseDiff as the base method, which limits the generality of the findings. Incorporating other representative approaches such as ESD, SalUN, and AC (Ablating Concepts in Text-to-Image Diffusion Models) would provide a more comprehensive and convincing demonstration of continual unlearning behavior across different unlearning paradigms.

4. While the paper mentions that continual unlearning may degrade the model’s general generative ability, the current evaluation mainly tests unrelated or random objects and styles to measure retention. A more informative evaluation would consider semantically related concepts to the forgotten target. For instance, when unlearning the concept “cat,” it would be more revealing to measure how well the model retains the ability to generate “tiger,” “lion,” or “leopard,” which are close in the embedding or visual space. Moreover, the paper lacks a broader assessment of general generation ability on a large-scale benchmark such as MS-COCO using metrics like CLIP Score or FID, which are standard in diffusion model evaluation.

---

> ### Author Response · Authors · 2025-11-25
> **Rebuttal Part 1 of 4**
>
> We thank the reviewer for their detailed feedback and for recognizing our work as a "clear problem definition of continual unlearning" with "practical plug-and-play remedies." Below, we address each concern in detail.
>
> ## (W1/Q1) Hyperparameter Sensitivity
> We thank the reviewer for raising this concern. To provide a more comprehensive analysis of each proposed add-on, we present a hyperparameter sensitivity study in the table below using Sculpting Memory on style erasure. Starting from our chosen hyperparameter values, we evaluate performance when each value is scaled by factors of 0.5 and 2.0. The table reports the mean performance along with standard deviation to quantify sensitivity to hyperparameter selection.
>
>
> We observe that our add-ons are generally robust to hyperparameter changes. For example, for the gradient projection and model merge methods, most standard deviations are lower than 3%.  Given this stability, we use a **fixed** set of L1/2 ($\lambda$=50), SelFT(top-k=5%), Gradient Projection(auxiliary concept number=200) for all experiments,  with sole exception of the model merging factor, which benefits from minor tuning.
>
>
> Notes that in the tables below, UA is the unlearn accuracy, RA-I is the in-domain retention accuracy, and RA-C is the cross-domain retention accuracy.
>
> **Model Merge**
> | Metric | Abstr(1) | Cartoon(3) | Ukiyoe(5) | Picasso(8) | Vibrant(12) |
> |--------|----------|------------|-----------|------------|-------------|
> | UA | 100.00% ± 0.00% | 100.00% ± 0.00% | 98.75% ± 1.77% | 97.66% ± 3.32% | 98.40% ± 2.28% |
> | RA-I | 85.35% ± 0.98% | 72.40% ± 2.22% | 51.57% ± 3.69% | 39.58% ± 1.47% | 28.99% ± 1.72% |
> | RA-C | 99.23% ± 0.25% | 99.17% ± 0.00% | 98.16% ± 0.85% | 98.12% ± 0.00% | 95.67% ± 0.37% |
>
>
> **Grad Proj**
> | Metric | Abstr(1) | Cartoon(3) | Ukiyoe(5) | Picasso(8) | Vibrant(12) |
> |--------|----------|------------|-----------|------------|-------------|
> | UA | 100.00% ± 0.00% | 100.00% ± 0.00% | 98.00% ± 2.83% | 98.91% ± 1.54% | 99.17% ± 0.00% |
> | RA-I | 98.44% ± 0.52% | 80.00% ± 5.60% | 67.92% ± 1.21% | 53.65% ± 3.11% | 37.44% ± 2.22% |
> | RA-C | 98.46% ± 0.27% | 98.67% ± 0.94% | 98.68% ± 0.42% | 97.06% ± 1.33% | 95.37% ± 0.22% |
>
>
>
>
> **SelFT**
> | Metric | Abstr(1) | Cartoon(3) | Ukiyoe(5) | Picasso(8) | Vibrant(12) |
> |--------|----------|------------|-----------|------------|-------------|
> | UA | 100.00% ± 0.00% | 99.59% ± 0.59% | 99.50% ± 0.71% | 99.54% ± 0.22% | 99.90% ± 0.15% |
> | RA-I | 92.29% ± 6.79% | 67.50% ± 5.66% | 40.42% ± 5.92% | 32.29% ± 2.05% | 14.59% ± 2.88% |
> | RA-C | 99.23% ± 0.00% | 98.61% ± 0.56% | 97.20% ± 3.54% | 98.50% ± 1.93% | 96.79% ± 3.07% |
>
>
> **L1**
> | Metric | Abstr(1) | Cartoon(3) | Ukiyoe(5) | Picasso(8) | Vibrant(12) |
> |--------|----------|------------|-----------|------------|-------------|
> | UA | 92.50% ± 10.61% | 95.00% ± 7.07% | 96.50% ± 4.24% | 97.82% ± 2.65% | 98.86% ± 0.16% |
> | RA-I | 95.83% ± 1.77% | 53.75% ± 13.56% | 33.96% ± 4.13% | 27.19% ± 1.03% | 20.21% ± 0.88% |
> | RA-C | 98.85% ± 0.01% | 99.34% ± 0.47% | 98.97% ± 0.00% | 98.50% ± 0.54% | 97.71% ± 2.66% |
>
> **L2**
>
> | Metric | Abstr(1) | Cartoon(3) | Ukiyoe(5) | Picasso(8) | Vibrant(12) |
> |--------|----------|------------|-----------|------------|-------------|
> | UA | 97.50% ± 3.54% | 95.83% ± 3.54% | 97.00% ± 0.71% | 97.03% ± 0.66% | 97.19% ± 0.59% |
> | RA-I | 96.98% ± 1.06% | 52.92% ± 5.89% | 37.19% ± 2.21% | 32.92% ± 1.18% | 28.34% ± 2.84% |
> | RA-C | 99.52% ± 0.14% | 98.83% ± 0.00% | 98.82% ± 0.00% | 98.00% ± 1.06% | 80.94% ± 1.68% |
>
> ## (Q2) Auxiliary Concept Selection
> For our gradient projection method, we utilize a set of prompts to build a text embedding subspace and then construct an orthogonal projection matrix to project the gradient. The set of prompts is randomly generated via an API call to GPT-4o-mini, which is prompted to generate random single-word concepts.
>
> As detailed in the 'Hyperparameter Sensitivity' section, we fix the number of auxiliary concepts at 200 in all main experiments, as this choice proves stable across various unlearning methods and concepts.
>
> Our sensitivity analysis in the section above (comparing 100, 200, and 400 concepts) confirms this robustness: the standard deviations are minimal ($\pm 0$ for UA, $\pm 2.2$ for RA-I, and $\pm 0.22$ for RA-C), indicating that performance is largely insensitive to the exact number of auxiliary concepts.

---

> ### Author Response · Authors · 2025-11-25
> **Rebuttal Part 2 of 4**
>
> ## (W2) Cost Analysis
>
> To provide a holistic cost analysis of baselines and our add-ons, we present both memory usage and runtime analysis in the table below.
>
> | Domain | Iterations | Metric | Base | L1 | L2 | SelFT | Merge | Projection |
> |--------|------------|--------|------|-----|-----|-------|-------|------------|
> | Style | 1000*N | Runtime (min) | 14.5 (N) | 0.2(N) | 0.2(N) | 2.6(N) | 0.1(N) | 1.38(N)  |
> | | | Storage (MB) | 73.1(1) | 0 | 0 | 0 | 73.1(N-2) | 0 |
> | Object | 2000*N | Runtime (min) | 29.6(N) | 0.2(N) | 0.2(N) | 2.6(N) | 0.1(N) | 1.38(N) |
> | | | Storage (MB) | 73.1(1) | 0 | 0 | 0 | 73.1(N-2) | 0 |
>
> In the provided table, we present runtime and storage analysis for Concept Ablation across both style and object erasure domains. Iterations denote the number of training iterations, and Base shows the number of minutes and megabytes incurred by the base unlearning method when used sequentially. The notation (N) indicates that cost grows linearly with the number of erasure requests, while (1) denotes constant cost. The columns L1, L2, Selective Fine-Tuning (SelFT), Merge, and Projection show additional costs relative to Base.
>
> All methods incur **small additional runtime costs**. Projection takes 1.38 minutes due to the LLM API call made to generate auxiliary concept captions for forming the text embedding subspace.
>
>
> All methods except Model Merge incur **no additional storage costs**. Model Merge requires access to all (N-1) previously trained models at the N-th request, therefore incurring an additional (N-2) storage cost relative to Base. However, the merging operation is very efficient and runs entirely on CPU.
>
> It is important to note that compared to simultaneous unlearning, sequential methods become significantly **scalable** as the number of erasure concepts increases. Simultaneous unlearning requires re-unlearning **all the concepts** from the base model at each request, incurring repeated computation that grows rapidly with scale. As shown in Figure x in the Appendix, our sequential unlearning maintains linear growth in training iterations, while simultaneous unlearning exhibits exponential growth. This exponential curve demonstrates that the number of required training iterations rapidly increases with each additional unlearning request, making sequential approaches substantially more efficient for multiple erasure concepts.

---

> ### Author Response · Authors · 2025-11-25
> **Rebuttal Part 3 of 4**
>
> ## (W3, Q3) More Unlearing Methods
> We thank the reviewer for this helpful suggestion. We would first like to clarify a possible misunderstanding: our evaluation does **not** rely solely on EraseDiff. In the original submission, we already include results for **Concept Ablation** (AC / Ablating Concepts in Text-to-Image Diffusion Models) as a main-text baseline, with **EraseDiff** reported in the appendix.
>
>
> To further strengthen the generality of our findings, we have conducted additional experiments along two axes:
>
> **Sculpting Memory (ICCV 2025, multi-concept erasure).**
>
> Following Reviewer 6PKG’s suggestion,  we conducted additional experiments on **Sculpting Memory** (ICCV 2025). We present full evaluation results for Sculpting Memory on our benchmark, along with all proposed add-ons (L1, L2, SelFT, Model Merge, and Gradient Projection) and found that continual unlearning remains a challenge even with recent state-of-the-art methods.
>
> As shown in Appendix Figure 12, Sculpting Memory significantly outperforms our previously benchmarked methods (ConAbl and EraseDiff). However, we note that the Sculpting Memory paper evaluates erasure with a maximum of 5 concepts, whereas our evaluation setting extends to 12 concepts. As shown in our results, while utility preservation starts strong, by the 12th concept both style and object retention experience significant degradation, with in-domain retention falling below 15% for both.
>
> Consistent with our findings on CA and EraseDiff, our gradient projection method achieves the strongest in-domain retention performance compared to all other add-ons (Appendix Figure 13). The improvement for object unlearning is particularly striking: our projection method achieves 87.50% in-domain retention accuracy and 82.08% cross-domain retention by the 12th concept, compared to just 10.62% and 5.08% respectively without our add-on.
>
>
> **ESD**
> To further address the reviewer's concerns regarding generalizability, we also provide celebrity erasure results using an additional unlearning method (ESD) and a larger model architecture (SDXL). As shown in the tables below, without any add-ons in sequential unlearning, ESD experiences utility collapse (single-digit retention accuracy) much more quickly (by the fourth celebrity) compared to ConAbl (Appendix Figure 14). Despite this, our proposed add-ons, especially Model Merge, greatly improve retention capability, with further gains achieved by combining Model Merge with our Gradient Projection method.
>
>
> **Sequential**
>
> | Metric | Neil Degrasse Tyson | Benicio Del Toro | Aziz Ansari | Oprah Winfrey | Betty White | Megan Fox |
> |--------|---------------------|------------------|-------------|---------------|-------------|-----------|
> | UA | 100.00% | 100.00% | 100.00% | 100.00% | 100.00% | 100.00% |
> | RA | 86.17% | 73.90% | 44.20% | 7.21% | 0.42% | 0.00% |
> | COCO-FID | 209.35 | 209.97 | 210.77 | 218.32 | 232.95 | 242.65 |
> | COCO-CLIP | 27.09 | 26.78 | 25.87 | 24.19 | 21.56 | 20.69 |
>
> **Model Merge**
> | Metric | Neil_Degrasse_Tyson | Benicio_Del_Toro | Aziz_Ansari | Oprah_Winfrey | Betty_White | Megan_Fox |
> |--------|---------------------|------------------|-------------|---------------|-------------|-----------|
> | UA | 100.00% | 100.00% | 98.14% | 95.31% | 97.32% | 95.52% |
> | RA | 86.17% | 87.91% | 86.18% | 67.99% | 57.12% | 50.62% |
> | COCO-FID | 209.35 | 209.39 | 209.09 | 209.12 | 209.40 | 210.16 |
> | COCO-CLIP | 27.09 | 27.20 | 27.17 | 27.07 | 26.99 | 26.92 |
>
> **Gradient Projection + Model Merge**
>
> | Metric | Neil_Degrasse_Tyson | Benicio_Del_Toro | Aziz_Ansari | Oprah_Winfrey | Betty_White | Megan_Fox |
> |--------|---------------------|------------------|-------------|---------------|-------------|-----------|
> | UA | 100.00% | 100.00% | 100.00% | 96.83% | 97.38% | 97.76% |
> | RA | 89.21% | 85.51% | 84.30% | 75.97% | 72.42% | 62.03% |
> | COCO-FID | 209.75 | 209.38 | 209.81 | 209.93 | 210.03 | 210.72 |
> | COCO-CLIP | 26.96 | 26.97 | 27.10 | 27.12 | 27.04 | 27.02 |
>
> Taken together, these new results demonstrate that (i) continual unlearning–induced utility collapse is a pervasive issue across **multiple, diverse unlearning paradigms** (CA, EraseDiff, Sculpting Memory, ESD), and (ii) our proposed add-ons act as **base-agnostic, plug-and-play** improvements, robustly enhancing retention across all methods and architectures we tested.

---

> ### Author Response · Authors · 2025-11-25
> **Rebuttal Part 4 of 4**
>
> ## (Q4) Evaluation Metrics
> We thank the reviewer for this thoughtful suggestion regarding broader evaluation metrics. We appreciate the opportunity to clarify that our original focus on classifier accuracy was intended to study continual learning with a more structured and controlled evaluation. But we agree that including FID and CLIP scores on MS-COCO provides a more holistic assessment. We explain our reasoning in detail below while also presenting new results that address this concern.
>
> We believe UnlearnCanvas provides a structured and controlled evaluation setting that offers certain advantages. The fine-tuned checkpoint accurately generates each style and object in the potential unlearning and retention sets, and the trained classifier achieves greater than 99% accuracy. As mentioned in UnlearnCanvas, classifier accuracy provides a more reliable evaluation since CLIP has been shown to be unstable and does not always accurately reflect erasure. Because CLIP considers the entire image, erasing "cat" from a prompt like "a cat on the beach" may remove the cat while the CLIP score remains high due to the background content still being generated correctly.
>
> That said, we appreciate the reviewer's point about the value of reporting common evaluation metrics to facilitate comparison with prior work. To address this, we conduct additional experiments on celebrity erasure (identity-based unlearning), which has been commonly evaluated in previous works. We select a sequence of 6 random celebrities to unlearn and an additional 6 for the held-out retention set. We employ the GIPHY celebrity classifier to measure unlearning accuracy (classifier error on unlearned celebrities) and retention accuracy (classifier accuracy on held-out celebrities). To evaluate general retention performance, we generate 5,000 images using MS-COCO prompts and report both FID and CLIP Score.
>
> As shown in the Appendix Figure 15, our findings remain consistent with those on UnlearnCanvas. Gradient Projection achieves the strongest in-domain retention, effectively preserving the model's ability to generate celebrities not targeted for removal. For cross-domain retention, measured via FID and CLIP Score on MS-COCO, Model Merge and SelFT demonstrate stronger performance. Importantly, we again observe that our methods can be combined for improved results: Gradient Projection paired with SelFT achieves the best overall performance (Appendix Figure 16).
>
> These additional CLIP Score and FID evaluations therefore further confirm that our add-ons provide **robust and consistent retention benefits** for unlearning methods in the continual setting, and we will include the full CLIP and FID results in the final revision.
>
>
> ## (Q4) Fine-grain Retention Evaluation
> We appreciate the reviewer’s suggestion on evaluating fine-grained retention for semantically related concepts. We agree that assessing how well the model retains concepts like “tiger,” “lion,” or “leopard” after unlearning “cat” is a very realistic and challenging setting.
>
> Our current work, however, is focused on a different but complementary aspect: whether continual unlearning leads to global utility degradation for existing methods, and how that happens. To this end, we intentionally build on UnlearnCanvas with **minimal** modifications. This controlled setup allows us to **isolate** the impact of continual unlearning  without introducing additional confounding factors.
>
> Even under the current, less challenging evaluation (without explicitly targeting semantic neighbors), we observe that existing unlearning methods already suffer **severe retention degradation** in the continual setting. We expect that introducing semantically related concepts as retention targets would only **amplify** these issues.
>
> Notably, our analysis already touches on semantic structure: as shown in Figure 7 and discussion in Section 7, we find that concepts closer in text-embedding space to the unlearning target are more susceptible to degradation, which motivated our gradient projection method.
>
> Thus, we believe that as unlearning methods progress to master continual unlearning in the existing benchmarks, evaluating fine-grained concept retention, as the reviewer suggests, will be a natural and important next step.

---

### Official Review · Reviewer_N1oH · 2025-11-01

**Soundness:** 2
**Presentation:** 3
**Contribution:** 3
**Rating:** 4
**Confidence:** 3

**Summary:**

Current machine unlearning methods typically assume that all deletion requests arrive simultaneously. However, in real-world scenarios, deletion requests are often sequential, a setting referred to as continual unlearning.
Existing approaches suffer from severe performance degradation under this setting, leading to both ineffective unlearning and the collapse of unrelated generation quality.
To address this, the authors systematically study continual unlearning for text-to-image diffusion models and propose three regularization-based strategies (update norm regularization, selective fine-tuning, and model merging), along with a semantic-aware unlearning method (gradient projection). These methods aim to mitigate parameter drift, improve retention of unrelated concepts, and minimize interference among semantically similar concepts.
Experimental results demonstrate that the regularization methods effectively alleviate the performance collapse problem, while the semantic-aware unlearning method achieves the most significant improvement in in-domain retention. Furthermore, it can be combined with other regularization techniques to achieve a better trade-off between unlearning effectiveness and image quality.

**Strengths:**

- Formally define and analyze continual unlearning in the text-to-image setting.
- Provides both theoretical and empirical insights into performance collapse due to parameter drift.
- Proposes modular regularization and semantic-aware techniques that can easily integrate with existing unlearning methods.
- Gradient Projection method effectively improves in-domain retention and reduces collateral forgetting.

**Weaknesses:**

- The study is limited to style and object deletions; it does not evaluate more practically relevant concepts such as NSFW, copyrighted, or identity-based content as previous works.
- All experiments are conducted on a single diffusion model within the UnlearnCanvas benchmark. The paper does not assess whether the proposed regularizers and gradient projection method generalize to other architectures or larger-scale diffusion models
- The benchmark setup relies on a limited base model and a relatively small, templated set of prompts for generation. This constrained setting—previously identified as a limitation of existing unlearning evaluation [1], which may not fully capture the diversity and complexity of real-world unlearning scenarios.
- While the proposed techniques (regularization, model merging, gradient projection) are well-motivated, they are largely adaptations or combinations of existing ideas.

[1] Ko, Myeongseob, et al. "Boosting alignment for post-unlearning text-to-image generative models." Advances in Neural Information Processing Systems 37 (2024): 85131-85154.

**Questions:**

In addition to the weaknesses,
- What's the computational overhead of different unlearning methods evaluated in the paper?

---

> ### Author Response · Authors · 2025-11-25
> **Rebuttal Part 1 of 3**
>
> We thank the reviewer for their thoughtful feedback and for recognizing that our work 'systematically' and '**formally defines and analyzes**' continual unlearning, providing '**both theoretical and empirical insights**.' We appreciate the constructive suggestions, which have helped us strengthen our evaluation. Below, we address each concern in detail.
>
> ## (W1) More Practical Setting
> As discussed in Section 4.2, we selected UnlearnCanvas as our primary evaluation benchmark due to its standardized and controlled experimental setting. The diffusion model checkpoint is fine-tuned on all styles and objects used for both unlearning and evaluation, achieving >99% accuracy with the trained classifier. This addresses limitations in previous works that relied on CLIP score, which has proven unstable and may not accurately reflect erasure success.
>
> As the first systematic study of continual unlearning for text-to-image diffusion models, we prioritized establishing a foundation **free from confounding variables** to fully focus on the continual unlearning. Robust evaluation metrics are essential for drawing conclusive claims about which methods genuinely improve utility preservation.
>
> Nevertheless, we still appreciate the reviewer pointing out the importance of evaluating on additional realistic scenarios. To demonstrate that our findings generalize beyond style and object erasure, we have conducted additional experiments on celebrity (identity-based) erasure using Concept Ablation as the base unlearning method. We evaluate all proposed add-ons from our paper, including L1 regularization, L2 regularization, Selective Fine-Tuning (SelFT), Model Merge, and Gradient Projection.
>
> Our results, shown in Appendix Figures [14,16], remain consistent with our findings on UnlearnCanvas. Gradient Projection achieves the strongest in-domain retention, effectively preserving the model's ability to generate other celebrities not targeted for removal. Importantly, we again observe that our methods can be combined for improved results: Gradient Projection paired with SelFT achieves the best overall retention performance.
>
> ## (W3) Prompt Diversity
>
> Thank you for the comment. In our experiments, we strictly follow the UnlearnCanvas benchmark's setting using the template "a {object} image in {style} style." While this limits prompt diversity, it ensures our results remain **comparable** to the benchmark without introducing *additional changes beyond* the continual setting. Notably, even in such a controlled setup, existing state-of-the-art unlearning methods already fail in the continual setting, highlighting the fundamental challenge it presents.
>
> That said, your concern regarding the prompt diversity is well received. Therefore, rather than modifying the UnlearnCanvas benchmark, we instead address the reviewer's concern by conducting celebrity erasure experiments with diverse evaluation prompts. To generate these prompts, we use GPT-4o-mini with the instruction: "Generate 50 unique captions for images containing {celebrity}. Each caption must include the word {celebrity} and be on its own line". This produces varied captions such as "Neil Degrasse Tyson inspires the next generation of scientists" and "Betty White teaches us to embrace life with humor and grace", rather than the standard template "a photo of {celebrity}". Despite this more challenging evaluation setting, unlearning accuracy across all proposed add-ons remains high.
>
> For additional experimental setup details, we select a sequence of 6 random celebrities to unlearn and an additional 6 for the held-out retention set. We employ the GIPHY celebrity classifier to measure unlearning accuracy (classifier error on unlearned celebrities) and retention accuracy (classifier accuracy on held-out celebrities). To further evaluate general retention performance, we generate 5,000 images using MS-COCO prompts and report both FID and CLIP Score.
>
> In this more challenging setting with diverse prompts that exposes weaknesses in methods prone to overfitting, all proposed add-ons still improve retention capabilities without sacrificing unlearning accuracy (Appendix Figure 15). The greatest performance gains come from combining our projection method with SelFT (Appendix Figure 16), demonstrating that our finding regarding the **combinability** of add-ons for enhanced performance generalizes to other erasure settings.

---

> ### Author Response · Authors · 2025-11-25
> **Rebuttal Part 2 of 3**
>
> ## (W2) Generalize to Other Architectures
> Thank you for the comment. The main reason we only evaluate on Stable Diffusion v1.4 is that most existing unlearning methods are proposed and implemented for this architecture.
>
>
> To our knowledge, adapting these unlearning methods to other architectures is non-trivial. As such, we chose to use their current implementations and focus on incorporating our add-ons on top.
>
> Given the limited rebuttal period, we explored an additional unlearning method, ESD (also per Reviewer 6PKG's request), which provides an official implementation on the larger SDXL architecture (our previously benchmarked methods do not provide SDXL implementations). As shown in the tables below, without any add-ons in sequential unlearning, ESD experiences utility collapse (single-digit retention accuracy) much more quickly (by the fourth celebrity) compared to ConAbl. Despite this, our proposed add-ons, especially Model Merge, greatly improve retention capability, with further gains achieved by combining Model Merge with our Gradient Projection method.
>
> **Sequential**
>
> | Metric | Neil Degrasse Tyson | Benicio Del Toro | Aziz Ansari | Oprah Winfrey | Betty White | Megan Fox |
> |--------|---------------------|------------------|-------------|---------------|-------------|-----------|
> | UA | 100.00% | 100.00% | 100.00% | 100.00% | 100.00% | 100.00% |
> | RA | 86.17% | 73.90% | 44.20% | 7.21% | 0.42% | 0.00% |
> | COCO-FID | 209.35 | 209.97 | 210.77 | 218.32 | 232.95 | 242.65 |
> | COCO-CLIP | 27.09 | 26.78 | 25.87 | 24.19 | 21.56 | 20.69 |
>
> **Model Merge**
> | Metric | Neil_Degrasse_Tyson | Benicio_Del_Toro | Aziz_Ansari | Oprah_Winfrey | Betty_White | Megan_Fox |
> |--------|---------------------|------------------|-------------|---------------|-------------|-----------|
> | UA | 100.00% | 100.00% | 98.14% | 95.31% | 97.32% | 95.52% |
> | RA | 86.17% | 87.91% | 86.18% | 67.99% | 57.12% | 50.62% |
> | COCO-FID | 209.35 | 209.39 | 209.09 | 209.12 | 209.40 | 210.16 |
> | COCO-CLIP | 27.09 | 27.20 | 27.17 | 27.07 | 26.99 | 26.92 |
>
> **Gradient Projection + Model Merge**
>
> | Metric | Neil_Degrasse_Tyson | Benicio_Del_Toro | Aziz_Ansari | Oprah_Winfrey | Betty_White | Megan_Fox |
> |--------|---------------------|------------------|-------------|---------------|-------------|-----------|
> | UA | 100.00% | 100.00% | 100.00% | 96.83% | 97.38% | 97.76% |
> | RA | 89.21% | 85.51% | 84.30% | 75.97% | 72.42% | 62.03% |
> | COCO-FID | 209.75 | 209.38 | 209.81 | 209.93 | 210.03 | 210.72 |
> | COCO-CLIP | 26.96 | 26.97 | 27.10 | 27.12 | 27.04 | 27.02 |
>
> ## (W4) Proposed Add-on Novelty
> Thank you for the comment. We want to reiterate that the primary goal of this paper is not to introduce entirely new methods, but to systematically study continual unlearning for text-to-image diffusion models and to understand why utility degradation occurs in this setting. Our findings show that the core problem lies in parameter drift and semantic confusion (Section 5.2, Section 7).
>
> This insight led us to investigate existing approaches that can potentially address these issues. While we recognized that choosing these existing methods may hurt our novelty, we humbly believe this is the most appropriate first step: **grounding our work in existing literature**. Specifically, we chose a diverse set of methods (L1, L2, Selective Fine-Tuning, Model Merge), each emphasizing different aspects of preventing drift.
>
> Moreover, we find that concepts closer in text embedding space to the unlearning target are more susceptible to utility degradation, which in turn motivates our semantic-aware gradient projection approach. While gradient projection alone is widely used in machine learning, in our humble opinion, its targeted application for preventing drift paired with semantically related concept sets is novel.

---

> ### Author Response · Authors · 2025-11-25
> **Rebuttal Part 3 of 3**
>
> ## (Q1) Cost Analysis
>
> To provide a holistic cost analysis of baselines and our add-ons, we present both memory usage and runtime analysis in the table below.
>
>
>
> | Domain | Iterations | Metric | Base | L1 | L2 | SelFT | Merge | Projection |
> |--------|------------|--------|------|-----|-----|-------|-------|------------|
> | Style | 1000*N | Runtime (min) | 14.5 (N) | 0.2(N) | 0.2(N) | 2.6(N) | 0.1(N) | 1.38(N)  |
> | | | Storage (MB) | 73.1(1) | 0 | 0 | 0 | 73.1(N-2) | 0 |
> | Object | 2000*N | Runtime (min) | 29.6(N) | 0.2(N) | 0.2(N) | 2.6(N) | 0.1(N) | 1.38(N) |
> | | | Storage (MB) | 73.1(1) | 0 | 0 | 0 | 73.1(N-2) | 0 |
>
> In the provided table, we present runtime and storage analysis for Concept Ablation across both style and object erasure domains. Iterations denote the number of training iterations, and Base shows the number of minutes and megabytes incurred by the base unlearning method when used sequentially. The notation (N) indicates that cost grows linearly with the number of erasure requests, while (1) denotes constant cost. The columns L1, L2, Selective Fine-Tuning (SelFT), Merge, and Projection show **additional costs** relative to Base.
>
> All methods incur **small additional runtime costs**. Projection takes 1.38 minutes due to the LLM API call made to generate auxiliary concept captions for forming the text embedding subspace.
>
>
> All methods except Model Merge incur **no additional storage costs**. Model Merge requires access to all (N-1) previously trained models at the N-th request, therefore incurring an additional (N-2) storage cost relative to Base. However, the merging operation is very efficient and runs entirely on CPU.
>
> It is important to note that compared to simultaneous unlearning, sequential methods become significantly **scalable** as the number of erasure concepts increases. Simultaneous unlearning requires re-unlearning **all the concepts** from the base model at each request, incurring repeated computation that grows rapidly with scale. As shown in Figure 25 in the Appendix, our sequential unlearning maintains linear growth in training iterations, while simultaneous unlearning exhibits exponential growth. This exponential curve demonstrates that the number of required training iterations rapidly increases with each additional unlearning request, making sequential approaches substantially more efficient for multiple erasure concepts.

---

> > ### Comment · Reviewer_N1oH · 2025-11-26
> >
> > Thank you for the detailed rebuttal. The explanation and the additional experiments addressed my questions. I will increase my score.

---

### Meta-Review · Area_Chair_wDcn · 2026-01-13

**Summary:**

Reviewers agree the paper identifies an important problem of continual/sequential unlearning and provides clear empirical evidence that existing diffusion unlearning methods collapse in retention due to cumulative parameter drift and semantic interference. However, the main concerns were (i) limited novelty, as the proposed regularizers/merging/projection are largely adaptations of known continual learning tools; (ii) limited evaluation realism and generality, since the original study focused on a controlled benchmark (UnlearnCanvas) with templated prompts and a single base model/architecture; (iii) also an incomplete characterization of hyperparameter sensitivity, computational/storage overhead, and comparisons/positioning against recent closely related multi-concept forgetting methods; and (iv) gaps in evaluation such as broader generation-quality metrics and more semantically fine-grained retention tests. Most of these are (partially) addressed by the authors on several points by adding cost analysis, sensitivity studies, broader metrics (COCO FID/CLIP), identity-based (celebrity) unlearning, and additional method/architecture coverage etc... which moves the balance toward acceptance.

**Reviewer Concerns:**

Few concerns remain outstanding or only partially addressed: while the rebuttal broadens evaluation to identity-based (celebrity erasure) unlearning, it still does not test more practically salient deletion categories explicitly raised by N1oH, such as NSFW or copyrighted content; the main experimental backbone continues to rely heavily on UnlearnCanvas with templated prompts, so realism is improved mainly through auxiliary experiments rather than the core benchmark;

Also novelty remains a lingering concern, as multiple reviewers view the methods as largely re-used ideas, and the rebuttal’s “systematic study + plug-in add-ons” framing does not fundamentally change that perception

**Reviewer Scores:**

The reviewers were mostly positive after the rebuttal and explicitely mentionned wilingness to increase their score and lean toward acceptance. Roughly, I would say that

- Reviewer N1oH (initial: 4 / marginally below accept): would likely increase to 5–6. They explicitly said the rebuttal and added experiments addressed questions and they will increase their score.

- Reviewer 6PKG (initial: 6 / marginally above accept): would likely stay at 6. Most of their concerns (sensitivity, cost, more methods, broader metrics) were directly addressed with added studies/experiments

-  Reviewer xbNs (initial: 6 / marginally above accept): would likely stay at 6, maybe slightly more since they replied that the in-depth analysis and experimental details addressed concerns

- Reviewer oE7J (initial: 6 / marginally above accept): would likely stay at 6. The rebuttal answers their main questions but the novelty concern is inherently unchanged, so a large jump is unlikely.

---

### Decision · Program_Chairs · 2026-01-26

Accept (Poster)